# Force-driven reversible liquid–gas phase transition mediated by elastic nanosponges

Keita Nomura [1], Hirotomo Nishihara [1], Masanori Yamamoto [1], Atsushi Gabe [1], Masashi Ito [2], Masanobu Uchimura [2], Yuta Nishina [3], Hideki Tanaka [4], Minoru T. Miyahara [5] & Takashi Kyotani [1]

Nano-confined spaces in nanoporous materials enable anomalous physicochemical phenomena. While most nanoporous materials including metal-organic frameworks are mechanically hard, graphene-based nanoporous materials possess significant elasticity and behave as nanosponges that enable the force-driven liquid–gas phase transition of guest molecules. In this work, we demonstrate force-driven liquid–gas phase transition mediated by nanosponges, which may be suitable in high-efficiency heat management. Compression and free-expansion of the nanosponge afford cooling upon evaporation and heating upon condensation, respectively, which are opposite to the force-driven solid–solid phase transition in shape-memory metals. The present mechanism can be applied to green refrigerants such as $H_2O$ and alcohols, and the available latent heat is at least as high as 192 kJ kg$^{-1}$. Cooling systems using such nanosponges can potentially achieve high coefficients of performance by decreasing the Young's modulus of the nanosponge.

[1] Institute of Multidisciplinary Research for Advanced Materials, Tohoku University, 2−1−1 Katahira, Aoba, Sendai 980−8577, Japan. [2] Advanced Materials and Processing Laboratory, Research Division, Nissan Motor Co., Ltd., 1 Natsushima-cho, Yokosuka, Kanagawa 237−8523, Japan. [3] Research Core for Interdisciplinary Sciences, Okayama University, Tsushimanaka, Kita-ku, Okayama 700-8530, Japan. [4] Research Initiative for Supra-Materials (RISM), Shinshu University, 4-17-1 Wakasato, Nagano 380-8553, Japan. [5] Department of Chemical Engineering, Kyoto University, Katsura, Nishikyo, Kyoto 615−8510, Japan. Correspondence and requests for materials should be addressed to H.N. (email: hirotomo.nishihara.b1@tohoku.ac.jp) or to H.T. (email: htanaka@shinshu-u.ac.jp)

Conventional cooling systems rely mostly on the bulk phase transition (BPT) of halocarbon-based refrigerants, which cause ozone depletion and global warming[1]. Recently, alternative systems have been extensively investigated by adopting the magnetocaloric effect[2–4], electrocaloric effect[5–7] and elastocaloric effect (eCE)[8–13]. eCE is especially promising because of its relatively large available latent heat (ca. 5–13 kJ kg$^{-1}$)[8,14,15], affording a coefficient of performance (COP) that is comparable (5–7)[11,13] to those of conventional BPT systems. However, the limitation of the latent heat based on the solid–solid phase transition hampers further improvement of COP. To develop more efficient cooling systems, it is important to explore alternative pathways for generating large latent heat with smaller external work. A probable method is to utilise physical adsorption or physisorption, the principle of which is shown in Fig. 1a. The van der Waals interactions between a solid surface and gas molecules (adsorbates) form a potential curve, and adsorbates are condensed on the solid surface usually through a multilayer adsorption mechanism. Inside the nanopores, two potential curves overlap to form a deeper potential (Fig. 1b), and adsorption occurs based on the mechanisms of micropore filling of pores <2 nm in size and capillary condensation in pores >2 nm at pressures much lower than the bulk equilibrium pressure. The density of the adsorbed refrigerant is close to the liquid density; therefore, adsorption corresponds to a phase transition[16]. Indeed, the heats of adsorption/desorption are comparable to the heats of condensation/evaporation of the bulk refrigerant. Thus, the adsorbed refrigerant is in different equilibrium conditions from that of the bulk. Figure 1c illustrates the two different equilibrium conditions of bulk and adsorbed refrigerant. The quasi-liquid adsorbed inside nanopores is in equilibrium with the gas phase, whose pressure is much lower than the bulk saturation pressure, and therefore, its gas–liquid equilibrium line can be considered as shifting downwards. For example, at point A in Fig. 1c, the bulk refrigerant is gaseous, whereas it is condensed as quasi-liquid in nanopores. If the adsorbed refrigerant at point A is forcibly squeezed out of a nanopore by mechanical deformation of the nanoporous material, the expelled refrigerant must obey the gas–liquid equilibrium of the bulk refrigerant (solid line in Fig. 1c), suggesting that the refrigerant at point A may evaporate immediately. Additionally, for nanoporous materials with sufficient softness and elasticity, the desorbed gas can be re-adsorbed upon shape recovery of the nanoporous materials. Thus, reversible phase transition can be expected on the mechanically flexible nanoporous materials by the combination of the existing physicochemical phenomena, adsorption and bulk phase transition. However, most conventional nanoporous materials like zeolites and metal-organic frameworks (MOFs) are too hard to be deformed significantly enough for squeezing the adsorbed refrigerants. Certain MOFs are considered soft or flexible, because they show reversible structure change[17–19], whereas their deformations are a kind of phase transition induced by the accommodation of guest molecules[17,18] or the irradiation of light[19], and their deformation by mechanical force is quite small[20]. Additionally, elastic porous materials with large pore sizes (>100 nm) like graphene aerogels[21–23] cannot be used because they lack the ability to adsorb gases. To realise the force-induced phase transition of refrigerants, nanoporous materials with sufficient elasticity and small nanopores (<10 nm), denoted as nanosponges, are necessary.

Our group has developed mechanically deformable microporous[24] and mesoporous[25] materials with sufficiently low bulk moduli (<1 GPa) by using single-layer graphenes as a major building component. In this work, we demonstrate that these elastic nanoporous materials can work as nanosponges that mediate the reversible liquid–gas phase transition by

mechanical force. Figure 1d, e schematically compare the phenomena occurring upon squeezing a plastic sponge absorbing liquid and nanosponge adsorbing quasi-liquid, respectively. While liquid is simply moved from the plastic sponge to outside (Fig. 1d), the liquid–gas phase transition occurs in the nanosponge (Fig. 1e).

## Results

**Phase transition of H$_2$O on zeolite-templated carbon.** First, zeolite-templated carbon (ZTC)[24,26], an elastic microporous material with a particle size of ~1–2 µm, was considered. We have recently reported a realistic structure model of ZTC, and revealed that ZTC comprises an ordered network of single-layer narrow graphene ribbon[27]. While the previous model consists only of carbon and hydrogen atoms, an improved model containing oxygen-functional groups (Fig. 2a) is constructed using the method recently developed[27] involving computer simulations[28–33], and it is used for calculating adsorption isotherms of H$_2$O, a polar adsorbate. The CHO ratio and the compositions of oxygen-functional groups were adjusted to those reported by experiment[26]. As a reference material, commercial activated carbon (AC) was used. Figure 2b shows X-ray diffraction (XRD) patterns of ZTC and AC. The sharp peak of ZTC at $2\theta = 6.3°$ ($d$-spacing of 1.4 nm) corresponds to its ordered structure derived from the (111) planes of the zeolite template (Fig. 2a)[27]. AC shows broad and weak peaks around $2\theta = 24°$ and 44°, corresponding to carbon 002 and 10 peaks, respectively. Because ZTC consists of nanoribbon-like single-layer graphene walls, its XRD pattern lacks the 002 and 10 peaks, which are derived from graphene stacking structures and large planer graphene structures, respectively. N$_2$ adsorption–desorption isotherms of ZTC and AC are shown in Fig. 2c. Both curves are classified as type-I isotherms by the IUPAC definition. The surface area of ZTC (2600 m$^2$ g$^{-1}$) is much higher than that of AC (1100 m$^2$ g$^{-1}$) due to its non-stacked graphene walls. The bulk modulus of ZTC determined by the stress–strain curve obtained with a mercury porosimeter (Supplementary Fig. 1)[24] is as small as 0.70 GPa. Compared to conventional nanoporous materials (Supplementary Table 1), ZTC is extraordinarily soft. The reversible deformation of ZTC is also confirmed by in situ observations via SEM (Fig. 3a). They show that the strain of ZTC is ~30%, and that the material recovers immediately after removing the force (Supplementary Movie 1), demonstrating sponge-like flexibility. According to Fig. 3a, the calculated Poisson's ratio of ZTC is approximately 0.29, which is between those of a plastic sponge (~0) and rubber (~0.49). By using the bulk modulus and the Poisson's ratio, the Young's modulus of a ZTC grain can be calculated as 0.88 GPa. In contrast, the tested AC sample is completely broken by the same pressing operation and exhibits no elastic deformation (Fig. 3b, Supplementary Movie 2). Such sponge-like elasticity has been difficult to achieve in other microporous materials including MOFs and organic-based materials; its achievement here is attributed to the excellent bendability of the single-layer graphene framework[24,27] as demonstrated by the ability to roll graphene into carbon nanotubes, while the Young's modulus of graphene under tensile deformation is very high (1.0–2.4 TPa)[34,35].

By using the elastic ZTC, the idea of the liquid–gas phase transition shown in Fig. 1e is indeed demonstrated by experiment with a home-made press chamber (Fig. 4a; more details are depicted in Supplementary Fig. 2). ZTC powder was mixed with 5 wt% polytetrafluoroethylene (PTFE) binder to form a sheet. Several such sheets were stacked and placed inside the closed chamber. Such system enables in situ monitoring of H$_2$O vapour pressure without (Fig. 4b) and with (Fig. 4c) pressing the sample.

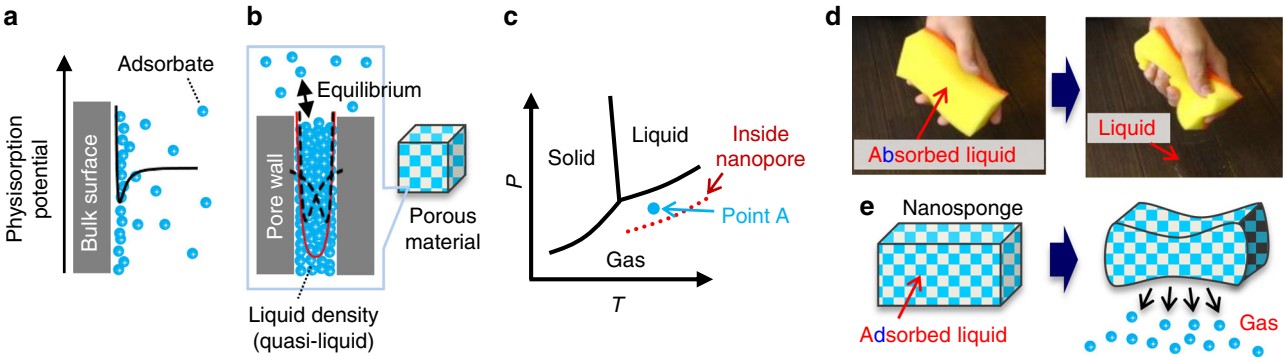

**Fig. 1** Adsorption and liquid–gas phase transition. **a** Physisorption potential (black line) on a bulk surface. Blue spheres indicate adsorbate molecules. **b** Physisorption potential in a nanopore. Two potential curves (black dotted lines) afforded by two pore walls are combined into a deeper potential curve (red line). **c** A simplified phase diagram of refrigerant ($H_2O$ is assumed). The liquid–gas equilibrium line shifts downwards in nanopores, as shown by the dashed red line. **d** Photos of a plastic sponge absorbing liquid (left) and its squeezed state (right). **e** Illustration of nanosponge adsorbing quasi-liquid (left) and its squeezed state (right)

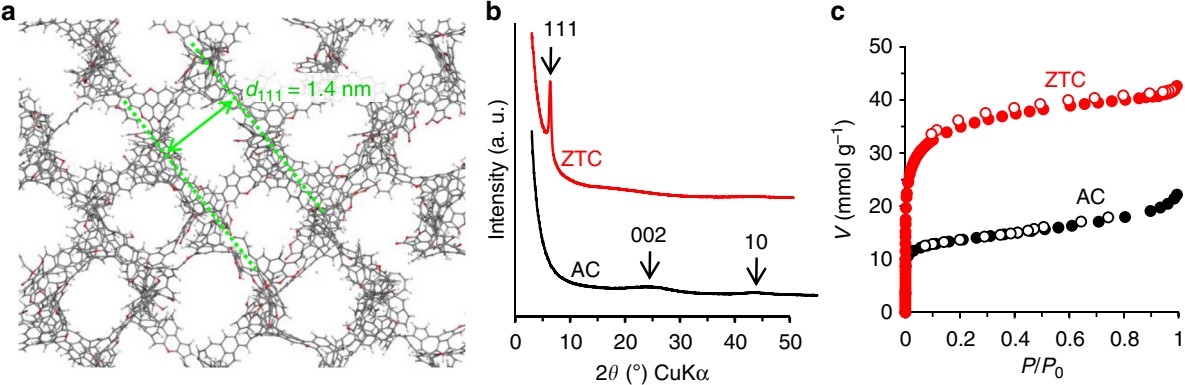

**Fig. 2** Properties of zeolite-templated carbon (ZTC) and activated carbon (AC). **a** Structural model of ZTC in which H, C and O are light grey, dark grey and red spheres, respectively. This model is constructed using the previously developed method[27]. **b** XRD patterns of ZTC (red line) and AC (black line). **c** $N_2$ adsorption–desorption isotherms of ZTC (red symbols) and AC (black symbols) measured at 77 K. Solid and blank symbols correspond to adsorption and desorption data, respectively

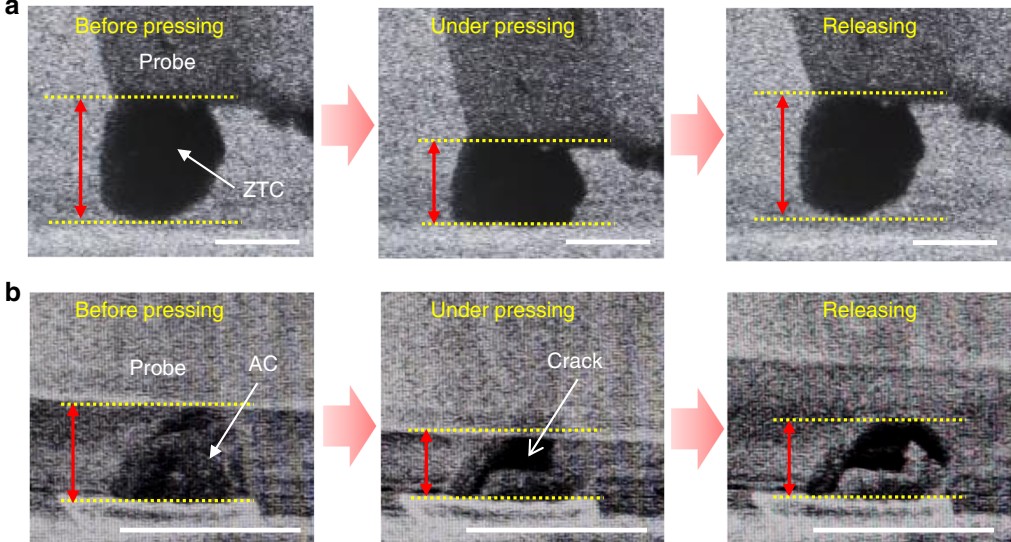

**Fig. 3** In situ SEM images upon compression of a sample particle. **a** ZTC. **b** AC. Left, middle, and right images depict before pressing, during pressing with a tungsten probe and after releasing the force, respectively. In each image, yellow dotted lines indicate the positions of the top and the bottom of the sample particle. Red arrows show the height of the sample particle. Scale bars are 1 μm. The corresponding movies are provided as Supplementary Movies 1 and 2

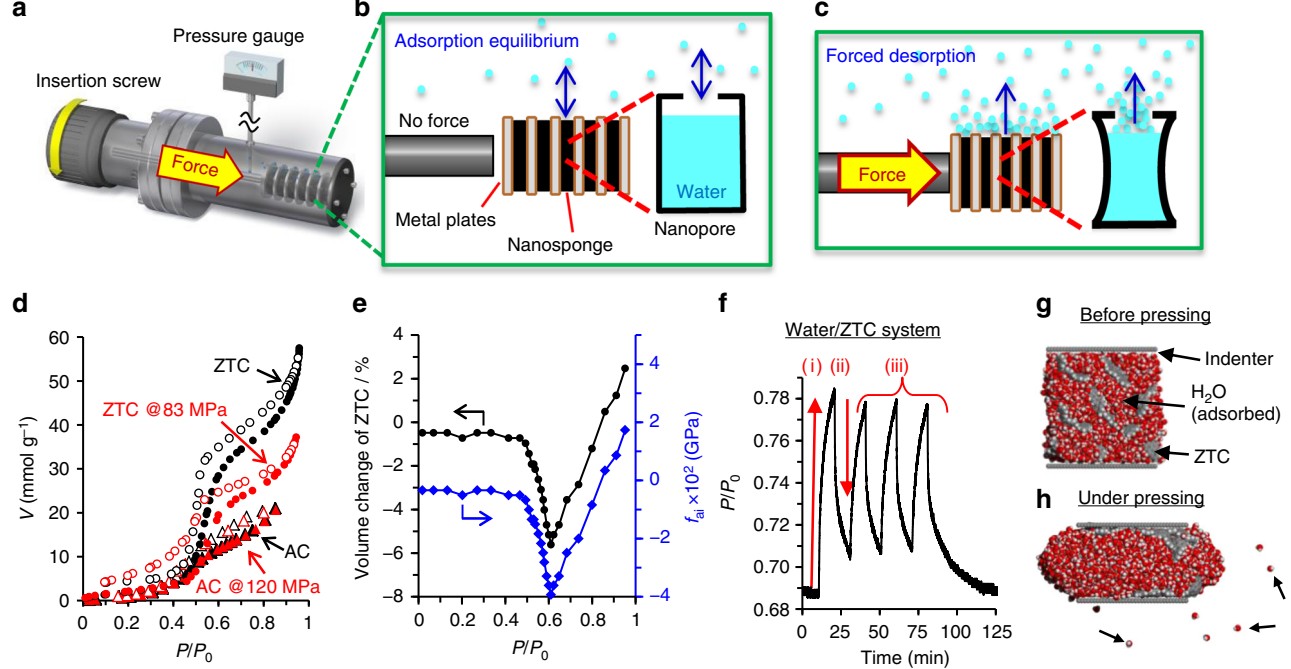

**Fig. 4** Force-driven liquid–gas phase transition of $H_2O$ mediated by ZTC. **a** Press chamber utilised for applying a mechanical force to the sample. It is connected to an automatic adsorption analyser equipped with a pressure gauge. By the insertion screw, it is possible to monitor the vapour pressure of the chamber **b** without and **c** with pressing the sample. **d** $H_2O$ vapour adsorption–desorption isotherms of ZTC (black symbols) and AC (red symbols) obtained during (triangle symbols)/without (circle symbols) pressing at 298 K. Solid and blank symbols correspond to adsorption and desorption data, respectively. **e** The volume change of ZTC induced by adsorption of $H_2O$ (black symbols and line), and the associated $f_{ai}$ (blue symbols and line). **f** Variations of the $H_2O$ vapour pressure ($P/P_0$) inside the press chamber observed in situ during applying/releasing a mechanical force to the ZTC specimen. (i): ZTC is pressed to release $H_2O$. (ii): ZTC is recovered to re-adsorb $H_2O$. (iii): Repeated release/adsorption cycles. **g**, **h** Snapshots of the MD simulation for forced $H_2O$ desorption from ZTC by compression. Red, dark grey and light grey spheres indicate oxygen, carbon, and hydrogen atoms, respectively. ZTC is filled with $H_2O$ (**g**) at 298 K, and compressed to 62% (**h**). Some of the desorbed molecules are indicated by arrows

In the beginning of the process, the $H_2O$ adsorption–desorption isotherms were recorded at 298 K with and without mechanical pressing at ~83 MPa (Fig. 4d). During pressing, the amount of adsorbed $H_2O$ is decreased by ~31% at $P/P_0 = 0.94$, because of the decreased pore volume caused by the ZTC deformation. Although the described mechanism appears very simple, the same process (pressing at ~120 MPa) applied to AC does not decrease its adsorption capacity (Fig. 4d), because of the lack of mechanical flexibility.

It is generally anticipated for gas adsorption in nanoporous materials that the narrower the pore becomes the lower the uptake pressure becomes. However, Fig. 4d does not demonstrate such effect, because of the peculiarity of $H_2O$ adsorption into carbon materials which are intrinsically hydrophobic. To obtain deeper understanding, grand canonical Monte Carlo simulations were carried out for $H_2O$-adsorption isotherms in the ZTC model containing oxygen-functional groups (Supplementary Fig. 3, Tables 2 and 3). The simulated isotherms with and without compression show very small difference for the $H_2O$ uptake pressure, indicating that $H_2O$ adsorption on ZTC is mainly governed by the strong dipole–dipole interaction between $H_2O$ and the oxygen-containing functional groups on ZTC and the effect of the increase of the London dispersion force by pore narrowing is less effective. Additionally, at the measurement shown in Fig. 4d, the sheet-shaped sample is sandwiched by metal plates and compressed, and therefore, diffusion of $H_2O$ vapour causes a kinetic problem, making the position of the $H_2O$ uptake not exactly the same as the equilibrium position calculated in Supplementary Fig. 3. Also, such diffusion issue affects the shape

of the hysteresis loop at low vapour pressure under compression of ZTC.

From the decrease in the pore volume (31%) at $P/P_0 = 0.94$ in Fig. 4d, the strain of ZTC can be calculated as 23% under pressing with 83 MPa, and the necessary work ($W_{ns}$) for the deformation of the ZTC, which adsorbs $H_2O$ is therefore calculated as 18.5 kJ $kg^{-1}$. $W_{ns}$ is the sum of the work ($W_b$) to deform blank ZTC and the additional work ($W_{ai}$) induced by $H_2O$ adsorption. When nanoporous materials accommodate adsorbate inside nanopores (when adsorption occurs), adsorbed molecules generates pressure working on nanoporous materials (adsorption-induced pressure, $f_{ai}$), and $f_{ai}$ causes deformation of nanoporous materials, which is known as adsorption-induced deformation[36–41]. To estimate $f_{ai}$, the adsorption-induced deformation of ZTC was measured as shown in Fig. 4e. Since ZTC possesses an ordered structure and shows a sharp XRD peak (Fig. 2b) unlike conventional porous carbon materials, it is possible to determine the degree of shrinkage/expansion upon the $H_2O$ adsorption by monitoring the change of the XRD peak position in situ during the $H_2O$ adsorption experiment. By $H_2O$ adsorption, the ZTC framework at first shrinks by 5.6% at $P/P_0 = 0.61$, whereas expands later on. At $P/P_0 = 0.95$, the ZTC framework eventually expands 2.5% compared to the original volume. Such a shrinkage-expansion behaviour is similar to those found in literature, but the magnitude of deformation is far greater as a microporous material[41], also demonstrating the remarkable softness of ZTC. From the volume change and the bulk modulus of ZTC, $f_{ai}$ can be obtained as shown in Fig. 4e. By using $f_{ai}$, $W_{ai}$ can be calculated according to Supplementary Eq. (20). For example at $P/P_0 = 0.91$,

$W_{ai}$ and $W_b$ are 4.8 and 13.7 kJ kg$^{-1}$, respectively, showing an unignorable effect of the adsorption-induced pressure.

Next, the phase transition of the adsorbed $H_2O$ into the gas phase, occurring under an applied mechanical force, is examined (Fig. 4f). For this purpose, $H_2O$ vapour is introduced to the chamber before the experiment to reach adsorption equilibrium at $P/P_0 = 0.69$ (Fig. 4b). After a mechanical stress of ~83 MPa is applied to the sheet (Fig. 4c), the relative pressure inside the chamber is increased to 0.79 [Fig. 4f, (i)]. Because the decrease in the dead volume of the chamber is negligible during pressing using the insertion screw (see Methods for the detailed calculation), the observed increase in $P/P_0$ can be attributed to the forced desorption of adsorbed $H_2O$ in accordance with the proposed phase transition mechanism shown in Fig. 1e. The feasibility of the proposed mechanism is further confirmed by molecular dynamics (MD) simulations. When ZTC including $H_2O$ inside its nanopores (Fig. 4g) is compressed, the liquid-density water is forcibly moved to outside of nanopores, and the expelled water is desorbed as gas (Fig. 4h). The corresponding movies showing the dynamic process at 298 and 350 K are available in Supplementary Movie 3 and 4, respectively, showing the enhanced desorption rate at a higher temperature. The MD simulation results reasonably explain the mechanism of the proposed phase transition by mechanical force. Moreover, to examine the effect of $H_2O$ inclusion in nanopores on the mechanical property of nanosponge, a simplified model using a carbon nanotube with a diameter of 1.36 nm was investigated by the MD simulation (Supplementary Fig. 4, Table 4 and 5). The calculated Young's modulus of the carbon nanotube (1.2 Ga) is close to that of ZTC (0.88 GPa). It is found that the $H_2O$ inclusion does not significantly affect the Young's modulus of the carbon nanotube. This is because water can easily escape from the carbon nanotube and the compression mode of water is different from hydrostatic pressing. The amount of $H_2O$ desorbed during the process of Fig. 4f, (i) is calculated as ~4.2 mmol g$^{-1}$, which almost agrees with the amount (5.1 mmol g$^{-1}$) estimated from Fig. 4d. Moreover, after releasing the mechanical force, the vapour pressure of $H_2O$ is decreased [Fig. 4f, (ii)], indicating the re-adsorption of the gas phase ($H_2O$ vapour) on the ZTC as it recovers its original porosity. Such force-induced desorption/adsorption cycles are repeated several times [Fig. 4f, (iii)], demonstrating that the reversible liquid–gas phase transition of $H_2O$ is possible because of the sponge-like deformation of ZTC. Although the phenomena itself is very simple and natural, this is the first demonstration of the reversible liquid–gas phase transition mediated by an elastic nanoporous material to the best of our knowledge.

**Phase transition of ethanol on graphene mesosponge.** The liquid–gas phase transition mediated by nanosponge is not limited to the $H_2O$/ZTC system, and many other combinations of refrigerants and nanosponge materials are possible. As a demonstration, another elastic nanosponge material of graphene mesosponge (GMS)[25] is employed together with ethanol as a refrigerant. The synthesis scheme of GMS is shown in Fig. 5a. $Al_2O_3$ nanoparticles are covered with extremely thin graphene sheets (approximately single-layer) by chemical vapour deposition (CVD) of $CH_4$ at 1173 K, followed by chemical etching with HF to remove the $Al_2O_3$ nanoparticles. The resulting porous carbon (carbon mesosponge) is then annealed at 2073 K to allow discrete graphene sheets to coalesce into continuous graphene walls. The GMS thus obtained mainly comprises single-layer graphene walls and exhibits very high porosity and flexibility[25]. As shown in Fig. 5b, adsorption amount of ethanol into GMS is significantly decreased by pressing (90 MPa), which is analogous

to the result of the $H_2O$/ZTC system (Fig. 4d). The adsorption amount at $P/P_0 = 0.92$ decreases to 58% under pressing. GMS deformation is calculated to be 36% from the total pore volume (2.79 cm$^3$ g$^{-1}$) and the true density (2.0 g cm$^{-3}$, general true density of carbon materials). Unlike the case of the $H_2O$/ZTC (Fig. 4d), the hysteresis loop closes at the same $P/P_0$ even under compression, indicating no serious problem on diffusion in ethanol desorption from mesoporous GMS. Figure 5c shows the result of the force-driven liquid–gas phase transition measurement performed in the same manner as that of the $H_2O$/ZTC system (Fig. 4f). The force-induced phase transition is also clearly observed in the ethanol/GMS system. Moreover, the response of desorption (pressure increase) and adsorption (pressure decrease) is faster compared to the $H_2O$/ZTC system, suggesting that the combination of nanosponge and refrigerant is important to achieve high rate phase transition.

Additionally, the mechanical stabilities of the two nanosponge materials (ZTC and GMS) are examined. Figure 5d shows $N_2$ adsorption isotherms of these samples before and after repeating pressing–releasing cycles for 100 times. In both samples, almost no porosity loss is found, indicating sufficiently high mechanical stabilities for repeating many pressing/releasing cycles.

**Potential for refrigeration.** The proposed phase transition has great potential for the design of a refrigeration system alternative to the conventional refrigeration based on the bulk phase transition (RBPT; Fig. 6a). In RBPT, an evaporator and a condenser are placed inside and outside of the target room, respectively. The heat $Q_L$ is removed from the room, while the heat $Q_H$ is released to the outside by the external work $W_{in}$ done by the compressor during the refrigeration cycle. The efficiency of a refrigeration system can be quantified by its COP, defined as the heat removed from the target room divided by the external work ($Q_L/W_{in}$). In the case of HFC-134a, a common halocarbon refrigerant, the operational pressure range is 0.34–1.2 MPa, and the corresponding temperature range is 277–323K, affording the COP around 5–7 (see Supplementary Table 6)[42]. A serious problem is that halocarbon-based refrigerants cause ozone depletion and global warming; for example, the global warming potential of HFC-134a is about 1300 times higher than that of $CO_2$[1]. Therefore, it is highly desirable to replace them with green media such as $H_2O$ or alcohols.

$H_2O$ and alcohols are interesting as refrigerant also from their large latent heat. For example, the latent heat of $H_2O$ is ~16 times greater than those of halocarbons. Hence, the use of $H_2O$ as a refrigerant can theoretically increase $Q_L$ by a factor of 16. However, $H_2O$ is also characterised by a relatively low vapour pressure (3.18 kPa at 298 K). When $H_2O$ is used in RBPT systems, its operational pressure ranges from 0.87 to 13 kPa (Supplementary Table 7)[43], which increases the volume of $H_2O$ vapour by ~2100 times (compared to that in conventional systems) and accordingly increases the value of $W_{in}$ by a factor of 20. As a result, the COP of the $H_2O$-based refrigerating system is reduced to that of a conventional halocarbon-based system, while significantly increasing its size. The relatively high operational temperature is another problem: in state **2** (Fig. 6a), its magnitude is ~504–537 K. The same problems exist also in alcohols. For these reasons, $H_2O$ and alcohols have never been used in RBPT systems, owing to the unchangeable $P-V-T$ relationship of their bulk phases. As shown above, the phase transition mediated by nanosponge is free from the $P-V-T$ relation of bulk refrigerants, and enabling the design of an alternative refrigeration system based on the mechanical force-induced phase transition of the adsorbate (RMPTA). While adsorption heat pumps (AHPs)[2] use adsorption in nanoporous materials, AHPs are categorised as

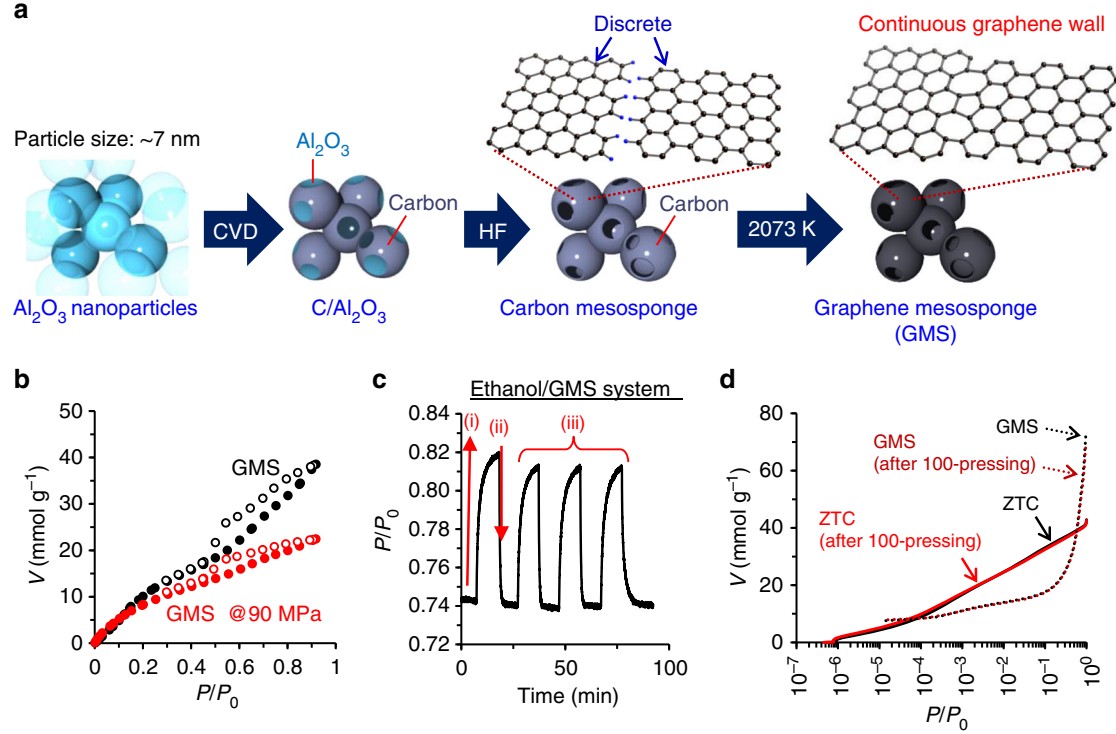

**Fig. 5** Graphene mesosponge (GMS) for phase transition of ethanol (EtOH). **a** Synthesis scheme of GMS. $Al_2O_3$ nanoparticles (particle size is ca. 7 nm) are uniformly covered with thin carbon layer by CVD. The carbon-coated $Al_2O_3$ (C/$Al_2O_3$) is washed with hydrofluoric acid (HF) to remove $Al_2O_3$. The resulting carbon mesosponge is annealed at 2073 K to obtain GMS. The carbon microstructures of carbon mesosponge and GMS are depicted as insets in which black and blue spheres indicate carbon and hydrogen atoms, respectively. **b** EtOH adsorption–desorption isotherms (298 K) on GMS with (red symbols) and without (black symbols) compressing at ~90 MPa. Solid and blank symbols correspond to adsorption and desorption data, respectively. **c** Variations of the EtOH vapour pressure ($P/P_0$) inside the press chamber observed in situ during applying/releasing a mechanical force to the GMS specimen. (i): GMS is pressed to release EtOH. (ii): GMS is recovered to re-adsorb EtOH. (iii): Repeated release/adsorption cycles. **d** $N_2$ adsorption isotherms (77 K) of ZTC and GMS before and after compressing them for 100 times at 83 MPa and 89 MPa, respectively. Solid and dotted lines correspond to the data of ZTC and GMS, respectively, while the data of pristine and compressed samples are shown by black and red colours

RBPT systems (Supplementary Fig. 5) and they differ intrinsically from the proposed RMPTA systems.

A prototype of the RMPTA system, depicted in Fig. 6b, consists of an evaporator and a condenser operated by the nanosponges. In the evaporator, the refrigerant adsorbed inside the nanosponge is forcibly desorbed (evaporated) under external work ($W_{ns}$), which corresponds to an endothermic process with the heat of desorption $Q_L$ (Fig. 6b, c). In the condenser, the nanosponge can be recovered via the free expansion accompanied by the exothermic adsorption of vapour with the heat of adsorption $Q_H$ (Fig. 6b, d). It is noteworthy that the correspondence of pressing/releasing to cooling/heating is opposite to that of eCE. When the nanosponge is not perfectly elastic, a part of $W_b$ is consumed by internal friction with the heat $Q_f$.

A system based on continuous refrigeration cycles is proposed in Fig. 6e, f (for more details, see Supplementary Fig. 6). Its condenser and evaporator represent closed chambers containing nanosponges and heat exchangers. These two chambers are connected through a vapour path equipped with a valve to avoid the reverse diffusion of vapour. In Fig. 6e, chamber A is used for recovering from the compressed state to adsorb vapour and release heat to the outside, while chamber B is compressed to desorb refrigerant for cooling the target room. When the adsorption (A) and desorption (B) processes are complete, the roles of these chambers are reversed by exchanging the two heat reservoirs (see Fig. 6f). At this time, the sensible heat $Q_{sh}$ is released to the target room. Thus, the COP of the RMPTA

prototype system can be expressed by the following equation (its detailed explanation is presented in Methods and Supplementary Fig. 6):

$$COP = \frac{|Q_L| - |Q_{sh}| - |Q_f|}{|W_{ns}|} \quad (1)$$

The COP of the prototype RMPTA system can be obtained by the Eq. (23) in Supplementary Methods. The necessary parameters can be obtained from adsorption isotherms of a refrigerant on a nanosponge measured at high ($T_H$) and low ($T_L$) temperatures, and a stress-strain curve of the nanosponge (Supplementary Fig. 7). For example, the COP for the $H_2O$/ ZTC system is 17.3 when $\Delta T$ ($= T_H - T_L$) is 5 K. In this system, 1 kg of ZTC adsorbs 0.9 kg of $H_2O$ at 298 K, and the amount of $H_2O$ that reversibly repeats phase transition [$w_{re}$ in Eq. (23)] is 0.15 kg among 0.9 kg. Thus, the net latent heat based on the mass of ZTC+ adsorbed $H_2O$ (1.9 kg) is 192 kJ kg$^{-1}$. The calculated COP is not very much high at $\Delta T = 5$ K; COP of RBPT is increased with decreasing $\Delta T$, and the COP of the HFC-134a system becomes 56.4 at $\Delta T = 5$ K. However, RMPTA based on the different mechanism from that of RBPT is intrinsically free from the restrictions of the conventional reverse Carnot cycle, and the COP can be continuously high regardless of $\Delta T$, as discussed in the following section. Moreover, COP can be potentially increased by decreasing the Young's modulus of nanosponge material as estimated in Fig. 6g.

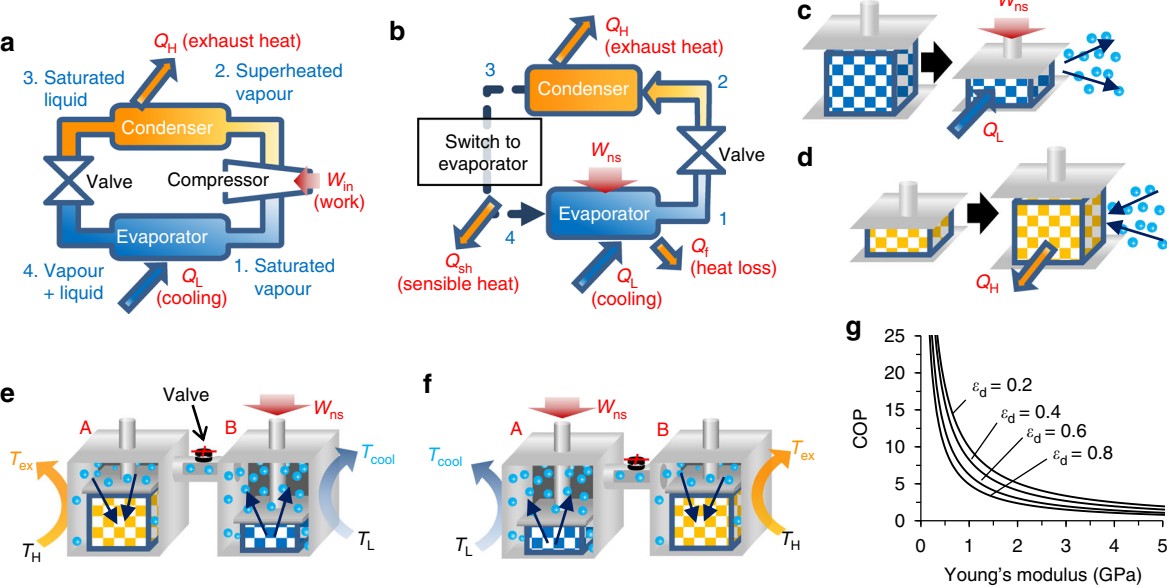

**Fig. 6** Comparison of the two refrigeration principles. **a** Schematic of the Refrigeration based on the Bulk Phase Transition (RBPT). The refrigerant undergoes four different phase transformations described by the numbers 1−4. Saturated vapour (1) is compressed with the external work ($W_{in}$) to be superheated vapour (2). 2 turns into saturated liquid (3) at the condenser by emission of exhaust heat ($Q_H$) to outside. 3 is expanded by the valve to be the mixture of vapour and liquid (4), and 4 is turned into 1 at the evaporator which is equipped with a heat-exchanger connected to a target room. The evaporator draws the heat ($Q_L$) from the target room. **b** Schematic of the Refrigeration based on the Mechanical-force-induced Phase-Transition of Adsorbate (RMPTA). Evaporator and condenser are equipped with nanosponges for phase transition and heat exchangers. 1 in the evaporator moved to the condenser to be 2. The valve is open only when the vapour pressure in the condenser is lower than that in the evaporator. When nanosponge in the condenser fully adsorbs refrigerant, the condenser is switched to the evaporator, and vice versa. The RMPTA system emits $Q_H$, sensible heat ($Q_{sh}$), and heat loss ($Q_f$), while $Q_L$ is drawn from the target room for cooling. **c, d** Illustrations of the **c** refrigerant desorption induced by the external work $W_{ns}$ and **d** refrigerant adsorption caused by the free expansion of the nanosponge. **e, f** A refrigeration system consisting of the two chambers (A and B) that are alternately compressed and released to serve as the evaporator and condenser, respectively. In **c–f**, blue spheres represent refrigerant vapour, while amber arrows indicate the moving directions of the vapour: desorption in **c**, B of **e** and A of **f**, adsorption in **d**, A of **e**, and B of **f**. **g** Approximate relations between the COP and Young's modulus of the nanosponge, which were obtained at various nanosponge strains $\varepsilon_d$ calculated using Eq. (35) when pore diameter ($d_p$) is 1.2 nm (the value in ZTC). Decreasing $\varepsilon_d$ increases the magnitude of COP and overall nanosponge volume, making the entire system bulkier

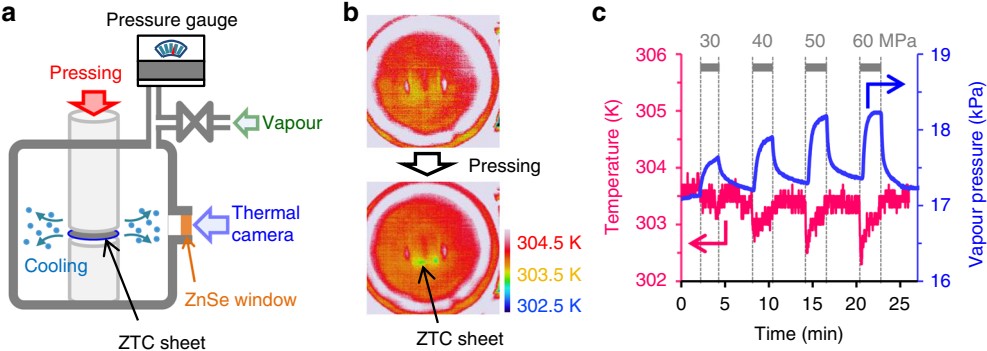

**Fig. 7** Temperature change by mechanical force-induced phase transition. **a** Experimental setup. A ZTC sheet is placed in a closed chamber filled with methanol vapour (17 kPa). The temperature of the ZTC sheet can be monitored by an infrared thermal camera through a ZnSe window. The chamber is connected to an automatic adsorption analyser for controlling and monitoring vapour pressure inside the chamber. **b** Snapshots of the infrared thermal camera without (top) and with (bottom) pressing the ZTC sheet. Red, yellow, blue colours correspond to 304.5, 303.5 and 302.5K, respectively. **c** The change of methanol vapour pressure (blue line) and temperature of ZTC (red line) when compression (30, 40, 50 and 60 MPa) and release are repeated

To examine the feasibility of the above idea, we developed a home-made device in which both the change of gas-phase pressure and the temperature of nanosponge can be monitored during the application of mechanical force on the nanosponge (Fig. 7a). For this experiment, a mechanically tough ZTC sheet was prepared by using graphene oxide (10 wt%) as an elastic binder. To obtain a fast adsorption/desorption response, methanol was selected as a refrigerant. At beginning, the ZTC sheet reached adsorption equilibrium under methanol vapour (17 kPa). Then, the change of temperature, as well as vapour pressure, was monitored with loading mechanical force of 30, 40, 50 and 60 MPa. As shown in Fig. 7b, the temperature change of the ZTC

sheet can be observed by an infrared thermal camera through a ZnSe window. The result of the in situ monitoring is shown in Fig. 7c. By loading mechanical force, vapour pressure increases, indicating the occurrence of the force-induced evaporation of methanol from ZTC. At the same time, the temperature of the ZTC sheet clearly decreases, demonstrating cooling by mechanical compression of ZTC adsorbing methanol. Moreover, the magnitude of cooling, as well as the increase of vapour pressure, becomes larger with increasing the mechanical force, i.e., along with the increase of the ZTC deformation. Figure 7c thus proves the most important concept of the RMPTA system: controlling temperature by phase transition generated by mechanical force applied to elastic nanosponge. On the other hand, when the external force is released, re-adsorption occurs, whereas the observed temperature rise was very small. This can be due to the measurement system which monitors the temperature change of only the edge of a ZTC sheet. When the ZTC sheet is pressed, desorption occurs mainly from the edge. On the other hand, when pressure is released, a small gap can be generated between the ZTC sheet and a pressing cylinder, and adsorption can occur the entire surface of the ZTC sheet. Thus, the temperature change of the sheet edge is less intense.

## Discussion

ZTC and GMS are actually not optimised for RMPTA, and appropriate material development might further improve the feasibility of RMPTA. Here the necessary requirements for such nanosponge materials are discussed. For this purpose, adsorption isotherms on a prospective nanosponge are considered (Supplementary Fig. 8), which is based on the temperature dependence of $H_2O$ vapour adsorption/desorption in mesoporous silicas[44]. It is predicted that a higher COP (~43) can be achieved by the development of nanosponge materials with large gaps between the adsorption amount on the pristine state at $T_H$ and that on the compressed state at $T_L$ [$w_{re}$, in the Supplementary Eq. (23)]. Unlike RBPT, such a high COP is not significantly decreased by the increase of $\Delta T$ (Supplementary Fig. 8). Moreover, it is expected that the value of COP can be further increased by decreasing the Young's modulus ($E$) of nanosponge. Figure 6g shows the approximate relation between COP and $E$ calculated by Supplementary Eq. (35) with taking the effect of adsorption-induced pressure into consideration (Supplementary Fig. 9a). COP can be basically increased by decreasing $E$, while the effect of the adsorption-induced pressure becomes remarkable when $E$ becomes very small (Supplementary Fig. 9b). Thus, the necessary requirements for the nanosponge materials to achieve a high COP are to have both large $w_{re}$ and small $E$. The effect of pore size ($d_p$) is also important but it is more complicated. The adsorption-induced pressure increases with increasing $d_p$, resulting in the decline of COP (Supplementary Fig. 9c). On the other hand, larger $d_p$ affords smaller $E$ and larger $w_{re}$, thereby increasing COP. Therefore, the pore size should be optimized with considering all these effects. As discussed above, there is still plenty of room to improve COP in the proposing system by optimization of nanosponge properties. A significant improvement of COP may be realised by the modification of graphene-based materials, such as ZTC or GMS, or development of new organic-based nanoporous materials such as MOFs, covalent-organic frameworks, and porous organic polymers.

In summary, force-driven reversible liquid–gas phase transition is demonstrated by mediating elastic nanoporous materials called nanosponges. The proposed phase transition enables the design of RMPTA systems operated by green refrigerants like $H_2O$ and alcohol, with great potential to achieve high COP depending on the amount of forced-desorption refrigerant and Young's

modulus of nanosponge materials. Potential efficiency improvements by the development of elastic nanoporous materials are significant, providing a new target for the study of nanoporous materials involving inorganic, organic, and metal-organic frameworks.

## Methods

**Materials**. ZTC was prepared by CVD[45]. Zeolite X (Union Showa, 13X) was placed in a vertical furnace and heated up to 873 K in $N_2$ flow. When the temperature reached 873K, $N_2$ flow was switched to a mixture gas of 15 vol% $C_2H_2$ and 85 vol% $N_2$, and CVD was performed for 4 h to deposit carbon inside the nanochannels of zeolite X. Subsequently, the gas flow was switched back to pure $N_2$, and the temperature was raised to 1123 K and kept for 3 h. After cooling down the sample to room temperature, zeolite X was dissolved by HF (47 wt%, Wako Pure Chemical Industries). Thus, ZTC was obtained. As a counterpart, a commercial activated carbon (AC; Shirasagi-P, Japan EnviroChemicals, Ltd.) was used. Graphene mesosponge (GMS) was synthesised according to the method we have developed[25]. Specifically, $Al_2O_3$ nanoparticles (Taimei Chemicals, TM300) were placed in a vertical furnace and heated up to 1173 K in Ar flow. When the temperature reached 1173 K, Ar flow was switched to a mixture gas of 20 vol% $CH_4$ and 80 vol% Ar, and CVD was performed for 2 h to deposit carbon on the entire surface of the $Al_2O_3$ nanoparticles. Then, the gas flow was switched back to pure Ar, and the sample was cooled down to room temperature. The resulting carbon-coated $Al_2O_3$ nanoparticles were immersed in HF (47 wt%) to remove the $Al_2O_3$ nanoparticles. The mesoporous carbon thus obtained was further annealed at 2073 K in Ar atmosphere (10 Pa) to obtain GMS. An mechanically tough ZTC sheet was prepared by mixing ZTC with graphene oxide (GO; NiSiNa materials). GO was prepared by the following method. Natural flake graphite (500 mg) was stirred in 95% $H_2SO_4$ (15 mL). $KMnO_4$ (1.5 g) was gradually added to the solution while keeping the temperature <283 K using a water bath. The mixture was then stirred at 308 K for 2 h. The resulting mixture was diluted with water (15 mL) under vigorous stirring and cooling so that the temperature did not exceed 323 K. The suspension was further treated with 30% aq. $H_2O_2$ (1.25 mL). The resulting suspension was purified by repeated centrifugation with water. The resulting graphite oxide was mechanically exfoliated, and single-layer graphene oxide was obtained from supernatant as dispersion after centrifugation at 3000 rpm for 5 min. Thus, GO suspension in water (0.3 wt%) was obtained. It was then mixed with ZTC by the following weight ratio, ZTC:GO = 9:1, and the mixture was filtered to form a uniform sheet.

**Characterisation**. The particle morphologies of ZTC and AC were observed by using SEM (Hitachi, S-4800). For in situ observation during the deformation of a single particle by loading mechanical force, a micro-manipulator for SEM (AD Science, PS4–2MM2LC-TH) was used. The tip of a tungsten probe (GGB Industries, PT-14-6705-B) was vertically cut by a Ga-focused ion beam apparatus (FEI, Helios NanoLab 600i) to make a flat surface. The sample was mounted on an Al substrate and pressed by the tungsten probe inside the SEM instrument to observe the deformation of a single particle in situ. Then, the probe was detached from the particle to observe the recovery. The ordered structures of the samples were characterised by XRD measurements with an X-ray diffractometer (Rigaku, MiniFlex600) with Cu Kα radiation. Textual properties of the samples were analysed by $N_2$ physisorption measurements at 77 K on the automatic adsorption analyser (Microtrac BEL, BELSORB-max). The specific surface areas were calculated by the Brunauer–Emmett–Teller (BET) method. The micropore volumes were calculated from the Dubinin–Radushkevich (DR) equation. The total pore volumes were calculated from the adsorbed amount at $P/P_0 = 0.96$. The stress–strain curve of ZTC under isotropic pressure was obtained by using a mercury porosimeter (Micromeritics Instrument Corporation, Autopore IV 9510). The sample powder was placed in a closed chamber, and Hg was introduced to the chamber under vacuum. Isotropic pressure ($P$ [MPa]) was then applied on the sample through the Hg while measuring its volume ($V_{Hg}$ [$m^3$]). At low pressure (<20 MPa), $V_{Hg}$ quickly decreases by Hg impregnation of the interparticle spaces. The bulk modulus ($K$ [MPa]) of the sample can be calculated from the linear part of the stress–strain curve at higher pressure (50–150 MPa), according to the following equation,

$$K = -V_0 \frac{\Delta P}{\Delta V_{Hg}} \tag{2}$$

where $V_0$ is initial sample volume [$m^3$].

In situ XRD measurement during $H_2O$ adsorption-isotherm measurement was carried out with a Rigaku SmartLab (XRD analyzer) with Cu Kα radiation. ZTC is placed in a closed chamber of the XRD analyser, and the chamber is connected to a BEL Japan BELSORP 18 (gas sorption analyser). These two apparatuses automatically communicate each other. When the gas sorption analyser records a datum at an equilibrium point, subsequently the XRD analyser begins a measurement. Thus, XRD patterns at each of adsorption data can be taken. From the shift of the peak around $2\theta = 6.3°$, volume contraction or expansion of ZTC is calculated.

**In situ $H_2O$-vapour adsorption measurements under pressing**. ZTC powder was mixed with PTFE (5 wt%) to form sample sheets. The sheets were dried at 423 K under vacuum and placed in a hand-made press chamber (Supplementary Fig. 2) connected to an automatic adsorption analyser (BEL Japan, BELSORB-max). To ensure the sample amount was sufficient for measurement, several sheets were sandwiched by thin stainless steel plates, and the sample stack was mounted in the press chamber. The press chamber was equipped with a rotary manipulator by which the sample stack could be mechanically pressed inside the closed chamber. In advance of the in situ measurement, the sample stack was pressed several times to stabilise the elastic deformation. Then, the relation between the torsion force loaded on the rotary manipulator and the force loaded on the sample sheets was obtained by using a ratchet with a torsion force meter (Kyoto Tool Co., GEK085-W36) and a compression tester (Shimadzu, AG-50kNXplus) equipped with a load cell. Using the relation obtained, the mechanical force loaded on the sample sheets was controlled by the torsion force loaded on the rotary manipulator.

Two types of in situ measurements were performed. First, adsorption–desorption isotherms with and without pressing were measured at 298 K. For comparison, the same in situ measurements were carried out also on AC. Second, the phase transition of adsorbed $H_2O$ into $H_2O$ vapour by mechanical pressing was demonstrated at 298 K. $H_2O$ vapour ($P/P_0 = 0.69$) was introduced into the press chamber for adsorption in ZTC at its original volume. The pressure inside the press chamber was monitored by the adsorption analyser. After reaching equilibrium, mechanical force (~80 MPa) was loaded on the sample stack as the vapour pressure of the chamber was monitored. After 10 min, the mechanical pressure was discharged and the sample was allowed to recover for 10 min. The pressing/releasing cycle was repeated four times. The decrease of the dead volume by this pressing operation was 0.13 cm$^3$, only 0.2% of the total dead volume (59 cm$^3$), corresponding to the increase of $P/P_0$ only by 0.0014. Thus, the increase of $P/P_0$ observed in Fig. 4f (ca. 0.9) was ascribed to $H_2O$ vapour generated by the forced desorption of adsorbed $H_2O$ by mechanical pressing.

**Measurement of in situ temperature change**. A ZTC sheet prepared by using GO as a binder was used as nanosponge. The sheet was dried at 423 K under vacuum and placed in a home-made chamber (Fig. 7a) connected to an automatic adsorption analyzer (BEL Japan, BELSORB-max) for controlling and monitoring vapour pressure. A thermal camera (Nippon Avionics Co., Ltd., Thermo FLEX F50) was used to monitor the temperature change of the ZTC sheet. At beginning, methanol vapour (17 kPa) was introduced into the chamber, and the ZTC sheet was achieved to adsorption equilibrium. Then, the ZTC sheet was compressed with a compression tester (Shimadzu, AG-50kNXplus) for 2 min, followed by releasing the force for 4 min. The pressing force was changed from 30 to 60 MPa by 10 MPa. The temperature of the experimental room was kept at 303 K.

## Data availability

The authors declare that the data supporting the findings of this study are available within the article and its Supplementary Information files. All data are available from the authors upon reasonable request.

## Code availability

Codes used for this work are available from the corresponding author upon reasonable request.

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

## Acknowledgements

The authors are thankful to Ms M. Ohwada and M. Ozawa for their experimental support. This work was supported by JST PRESTO Grant no. JPMJPR1317; JST PREST network; JST CREST Grant no. JPMJCR1324 and JPMJCR18R3; JSPS KAKENHI Grant Number 16J06543, 17H01042 and 17H03097; the Dynamic Alliance for Open Innovation Bridging Human, Environment and Materials; and the Network Joint Research Centre for Materials and Devices.

## Author contributions

K.N. synthesised ZTC and GMS, and carried out adsorption measurements in situ during mechanical pressing. H.N. designed the RMPTA theory and the relevant experiments. M.Y. performed necessary experiments for estimating the COP of the $H_2O$/ZTC system. A.G. optimised the preparation conditions of elastic ZTC/GO composite sheets. M.I. contributed to the design of the RMPTA for automobile application. M.U. contributed to the design of RMPTA applicable to practical devices. Y.N. has synthesised GO and developed the technique to form uniform membranes. H.T. simulated the $H_2O$ adsorption isotherms on ZTC and also demonstrated the forced desorption of $H_2O$ by the MD simulation. H.T. also examined the effect of force-induced deformation of ZTC. M.T.M. supervised the simulation done by H.T. T.K. supervised the whole project and analysed the mechanical properties of ZTC.

## Additional information

**Competing interests:** The authors declare no competing interests.

