## [Peer Review File · Nature Communications]

Reviewers' comments:

Reviewer #1 (Remarks to the Author):

The paper reports a prototype of a new refrigeration device, based on the vapor-liquid phase transitions in an elastic nanoporous medium. The chosen medium (graphene-based nanosponges) has small pore sizes, high surface area and low elastic moduli. A combination of these parameters provides a opportunity to induce the phase transition by a mechanical load.

This idea clearly shows a viable alternative to the conventional vapor-compression cycle. The proposed cycle does not require the adsorbate to have thermodynamic properties inherent to CFCs, and can operate with green refrigerants, such as water. In addition to explaining the principle and fabricating the device, the authors supported their work with a molecular simulation model, which sheds light on the underlying molecular mechanism.

The idea is new, creative and original. It is likely to have an impact not only on the development of new refrigeration technologies, but also on the development of new nanoporous materials with tailored elastic properties, as well as the theoretical understanding of coupling adsorption with mechanics. I believe, the paper will make a significant impact.

In addition to scientific impact, it could have a broader educational impact, since it brings a creative refrigeration cycle example, which could be explained to a non-expert. I would use this example to illustrate new refrigeration technologies when teaching the undergraduate thermodynamics course.

It looks like the authors provided enough details to reproduce both experimental and simulation parts of this study.

Overall, I like the paper a lot, and would be delighted to see it published. It is very nicely written, and I do not have any major issues with it. There are a couple of rather minor issues, which I would like the authors to discuss.

The phenomenon presented in the paper is inherently related to another phenomenon, which has attracted a lot of attention recently: adsorption-induced deformation. The authors do not discuss it, and neglect it in their COP estimates. However, the lower is the elastic modulus of the porous

material, the more pronounced the adsorption-induced deformation is. It seems to me that while at reasonably high Young's moduli (e.g. 0.7 GPa considered in the paper), the adsorption-induced strains will remain negligible, at lower moduli it may counteract the strains due to applied external stresses. Probably, the inclusion of the correction for adsorption-induced deformation in the estimate for the COP, could limit the infinite increase of COP at low Young's modulus shown in Figure 6(g).

When significant mechanical load is applied to a fluid-saturated porous medium, the response depends on the elastic properties of both constituents: solid and fluid. Isn't the fluid compressibility another parameter, which determines the COP? Especially that the fluid compressibility in nanopores is known to be different as compared to the macropores.

Technical comments/typos:

Line 38: "is equivalent" -> "is close"

Line 115: "mechanical force of 83 MPa" -> "mechanical stress of 83 MPa"

Line 220: "imaginary nanosponge" -> "prospective nanosponge"

Figure 4(e): can you plot the force as a function of time on the same plot?

Supplementary Table 3: "...their Young's moduli are not available in literature" The moduli (pore-load moduli) of nanoporous materials can be calculated from the adsorption-induced strains, and related to the conventional Young's moduli. This has been done in a number of works recently.

Reviewer #2 (Remarks to the Author):

Report on the Paper:
**Force-driven reversible liquid-gas phase transition mediated by elastic
 nanosponges**

by K. Nomura *et al.*

The paper deals with “graphene-based” microporous materials which are able to adsorb (physisorption) and desorb an adsorbate – water and alcohols – under expansion and compression of the material. Van der Waals interactions combined with nano-confinement in the pores lead to a high density – liquid-like, the authors say – adsorbate multilayer. The corresponding adsorption/desorption heats are found comparable with bulk phase condensation/evaporation of conventional refrigerants. The idea is to exploit the “soft” elasticity of the proposed material – a frame of graphene nano-ribbons with low bending rigidity – to compress and expand the material through an externally applied force thereby respectively forcing de-adsorption (evaporation) and re-adsorption (condensation) of the adsorbate.

The authors discuss the structure of the micro/mesoporous material as obtained by molecular dynamics simulations, x-ray diffraction curves and Nitrogen adsorption data for their zeolite-template carbon (ZTC) in comparison with activated carbon (AC). They also illustrate the elasticity of ZTC in comparison with the rigidity and brittleness of AC with SEM micrographs (supplementary movies available) of a microparticle of both materials compressed by a probe to demonstrate the “nanosponge” behavior of ZTC.

The central points of the paper are figure 4 and 5, where the authors sketch their device and measure the adsorption/desorption isotherms under compression of the micro/mesoporous material (beside the microporous ZTC and water, they also discuss the mesoporous graphene mesosponge (GMS) using ethanol as adsorbate). The authors also show the stability of the process by a few compression-decompression cycles of their materials. Finally a scheme is presented for a newly conceived refrigeration cycle using the force-induced desorption mechanism to pump heat from a low to a high temperature chamber.

Although eventually understandable, the manuscript forces the reader to repeatedly shift back and forth between main text and Supplementary Information, which hampers the effective communication of the most important results. A certain familiarity with porous materials is required of the reader, which could make the text poorly readable by non-specialists. The central idea of the paper seems interesting from the point of view of potential applications. I have however substantial perplexities as concerning the overall focus of the paper and the novelty of the technical results, see the list of comments below:

- The materials the authors exploit have already been published.
- Most of the techniques used to quantify the claims made in the paper are standard.

- Simulations are also already published. They are never clearly explained in the text and seem to be used mainly for illustrative purposes.
- As a contradiction, a very long section is devoted to simulations in the Supplementary Information.
- As a major issue, the authors do not provide a clear cut interpretation of their measurements. One may assume that what they observe is, in its main lines, related to the change in pore size (and geometry?) under compression. Now, if I am not misled, the effect of pore size on adsorption is treated already in the specialized literature. The question would rather be to explain the hysteresis loops in the adsorption/desorption isotherms. How is this related to the geometrical modification induced by the strain is however not clarified in the paper.
- Overall, the main purpose of the paper is left somewhat vague. Do the authors wish to contribute fundamental understanding or do they aim at proposing a new application in thermal sciences? In the former case they should focus more on the explanation of the phenomenology. If, on the other hand, their contribution is targeted to the application of soft nanosponges to heat management, they should demonstrate their cooling system in practice, e.g., by providing a proof of concept where heat transfer is actually measured.

In conclusion, I am not convinced the paper is appropriate for Nature Communications and, after revision, it would most probably find a better collocation in a specialized journal devoted to applications of nanomaterials.

Reviewer #3 (Remarks to the Author):

The authors report experiments and phenomenological considerations on a force-driven liquid-gas phase transition in mechanically deformable nanoporous materials. Moreover, it is suggested to use this effect for refrigeration with water and alcohols as “green” working fluids. Unfortunately, the study has a number of shortcomings regarding the presentation and interpretation of the experiments. Moreover, it does not contain any new mechanistic insights and remains on a rather hypothetical level with regard to the application in refrigeration. Therefore, unfortunately, I do not see the horsepower of this paper which would make it suitable for a publication in Nature Communications. In the following a few more detailed remarks:

(1) Line 119: It is not clear to me what the authors mean with “new mechanism of phase transition”? Because of the fore-induced compression the pore space cannot accommodate the condensed molecules in the pores. Thus, they are expelled, do not experience the attractive fluid-solid interactions and evaporate in a classical bulk manner. Thus, I do not see a novel mechanism here.

(2) The authors do not consider in their thermodynamic considerations the tensile Laplace pressure and sorption stresses in the confined liquids and nanopores respectively. In particular in the capillary-condensed state with concave menisci between liquid and vapour the Kelvin equation in combination with the Young-Laplace equation dictates enormous negative (tensile) pressures in the fluid for the mesoporous sponges. They result in substantial deformations, in particular contractions of nanoporous materials upon capillary condensation even without the application of external pressures (see G. Gor et al. Applied Physics Reviews 4, 011303 (2017) and references therein).

(3) In the case of the application of soft porous materials with low Young modulus, suggested by the authors as a way to increase the coefficient of performance, the sorption-deformations mentioned above will result in substantial nanopore contractions and thus in a spontaneous fluid-adsorption-induced reduction of pore space - even without external forces. Thus, the overall porosity available for fluid adsorption will be substantially reduced which may even overbalance the increased contraction response to external pressure and thus decrease the COP.

(4) The authors discuss the refrigeration cycles theoretically only, but do not show any experimental data actually proofing their cooling/heating concept.

Response to the reviewer #1

Thank you very much for your precious comment. We have revised the manuscript in accordance with your opinions as shown below.

Your specific comment:

1. The phenomenon presented in the paper is inherently related to another phenomenon, which has attracted a lot of attention recently: adsorption-induced deformation. The authors do not discuss it, and neglect it in their COP estimates. However, the lower is the elastic modulus of the porous material, the more pronounced the adsorption-induced deformation is. It seems to me that while at reasonably high Young's moduli (e.g. 0.7 GPa considered in the paper), the adsorption-induced strains will remain negligible, at lower moduli it may counteract the strains due to applied external stresses. Probably, the inclusion of the correction for adsorption-induced deformation in the estimate for the COP, could limit the infinite increase of COP at low Young's modulus shown in Figure 6(g).

Response:

Thanks for your suggestion. As you pointed out, when nanoporous materials accommodate adsorbate inside nanopores (when adsorption occurs), adsorbed molecules generates pressure working on nanoporous materials (adsorption-induced pressure, f_{ai}), and f_{ai} causes deformation of nanoporous materials (adsorption-induced deformation). Regarding this phenomenon, the following two effects should have been considered for the COP calculation. One of them is the change of adsorption capacity caused by the adsorption-induced deformation. This effect is already reflected to adsorption isotherm data, and the present COP estimation is based on the adsorption amount from the isotherm data. So, there is no need to consider about this effect. The other is the effect of f_{ai} on the change of the total work (W_{ns}) applied for forcible desorption in the equation (14).

$$\begin{aligned} \text{COP} &= \frac{|Q_L| - |Q_{sh}| - |Q_f|}{|W_{ns}|} \\ &= \frac{w_{re}\Delta_{vap}H - (T_H - T_L)(c_{re}w_{re-ads} + c_{ns}w_{ns}) - |Q_f|}{|W_{ns}|} \end{aligned} \quad (14)$$

W_{ns} is actually the sum of the work which is necessary to deform blank nanosponge (W_b) and the work against the force derived from adsorption-induced pressure (W_{ai}). Therefore, we have added the following equation in the revised version (Supplementary method):

$$W_{ns} = W_b + W_{ai} \quad (18)$$

W_b is expressed as follows:

$$W_b = \frac{1}{2}Fx = \frac{1}{2}\frac{S}{L_0}Ex^2 = \frac{1}{2}SL_0E\varepsilon_d^2 = \frac{1}{2}V_0E\varepsilon_d^2 \quad (19)$$

where V_0 is the initial volume of nanosponge [m³].

W_{ai} is described as follows:

$$W_{ai} = V_0\varepsilon_d f_{ai} \quad (20)$$

When a nanoporous material expands by adsorption, f_{ai} is described as follows (Gor, G. Y. *et al.*, *Langmuir* **26** (2010) 13021):

$$f_{ai} = f_s - f_0 = -\frac{2\gamma_{sl}}{d_p} + \frac{R_g T \rho_{re}}{M_{re}} \ln\left(\frac{P}{P_0}\right) + (P_0 - P) - f_0 \quad (21)$$

where f_s [Pa], f_0 [Pa], γ_{sl} [N m⁻¹], d_p [m], ρ_{re} [kg m⁻³], R_g , M_{re} [kg mol⁻¹] are solvation pressure, a prestress at the initial sample volume V_0 , the pore wall-liquid surface tension, pore diameter of nanosponge, the density of the adsorbed refrigerant, gas constant, molecular weight of adsorbed refrigerant, respectively. The directions of f_s and f_0 are opposite, and f_0 has a minus value. From equations (20) and (21), W_{aid} is described as follows:

$$W_{ai} = V_0\varepsilon_d \left(-\frac{2\gamma_{sl}}{d_p} + \frac{R_g T \rho_{re}}{M_{re}} \ln\left(\frac{P}{P_0}\right) + (P_0 - P) - f_0 \right) \quad (22)$$

For the estimation of the relation between COP and Young's modulus, it is necessary to obtain f_{ai} . Thus, we have measured the degree of adsorption-induced deformation by *in situ* XRD during H₂O adsorption isotherm measurement, and determined f_{ai} . The results have been added as Fig. 4e in the revised manuscript as shown below.

Figure 4e The volume change of ZTC induced by adsorption of H₂O, and the associated f_{ai} .

Since zeolite-templated carbon (ZTC) possesses an ordered structure and shows a sharp XRD peak, it is possible to know the degree of shrinkage/expansion upon the H₂O adsorption by monitoring the change of the XRD peak position. By H₂O adsorption, the ZTC framework at first shrinks by 5.6% at $P/P_0 = 0.61$, whereas expands later on. At $P/P_0 = 0.95$, the ZTC framework eventually expands 2.5% compared to the original volume. Such a shrinkage-expansion behavior is similar to those found in literature, but the magnitude of deformation is far greater as a microporous material, also demonstrating the remarkable softness of ZTC. From the volume change and the bulk modulus of ZTC, f_{ai} can be obtained as shown in Fig. 4e. By using f_{ai} , W_{ai} can be calculated according to the above equation (20). For example at $P/P_0 = 0.91$, W_{ai} and W_b are 4.8 kJ kg⁻¹ and 13.7 kJ kg⁻¹, respectively, showing an unignorable effect of the adsorption-induced pressure.

Thus, we have added the following discussion regarding the adsorption-induced deformation in the main text in page 8:

From the decrease in the pore volume (31%) at $P/P_0 = 0.94$ in Fig. 4d, the strain of ZTC can be calculated as 23% under pressing with 83 MPa, and the necessary work (W_{ns}) for the deformation of the ZTC which adsorbs H₂O is therefore calculated as 18.5 kJ kg⁻¹. W_{ns} is the sum of the work (W_b) to deform blank ZTC and the additional work (W_{ai}) induced by H₂O adsorption. When nanoporous materials accommodate adsorbate inside nanopores (when adsorption occurs), adsorbed molecules generates pressure working on nanoporous materials (adsorption-induced pressure, f_{ai}), and f_{ai} causes deformation of nanoporous materials, which is known as adsorption-induced deformation³⁶⁻⁴¹. To estimate f_{ai} , the adsorption-induced deformation of ZTC was measured as shown in Fig. 4e. Since ZTC possesses an ordered structure and shows a sharp XRD peak (Fig. 2b) unlike conventional porous carbon materials, it is possible to determine the degree of shrinkage/expansion upon the H₂O adsorption by monitoring the change of the XRD peak position in situ during the H₂O adsorption experiment. By H₂O adsorption, the ZTC framework at first shrinks by 5.6% at $P/P_0 = 0.61$, whereas expands later on. At $P/P_0 = 0.95$, the ZTC framework eventually expands 2.5% compared to the original volume. Such a shrinkage-expansion behavior is similar to those found in literature, but the magnitude of deformation is far greater as a microporous material⁴¹, also demonstrating the remarkable softness of ZTC. From the volume change and the bulk modulus of ZTC, f_{ai} can be obtained as shown in Fig. 4e. By using f_{ai} , W_{ai} can be calculated according to Supplementary equation (20). For example at $P/P_0 = 0.91$, W_{ai} and W_b are 4.8 kJ kg⁻¹ and 13.7 kJ kg⁻¹, respectively, showing an unignorable effect of the adsorption-induced pressure.

Moreover, we have thoroughly revised the calculation of COP to include the effect of the

adsorption-induced pressure, in the Supplementary Method. The major results are summarized in the following Figure A1.

Figure A1 The relation between COP and Young's modulus. (a,b) The data shown in the original manuscript. (c,d) The revised data considering the effect of the adsorption-induced pressure. (b) and (d) show the small Young's modulus regions of (a) and (c), respectively.

Figure A1a is the graph which was shown in the original manuscript. The consideration of the adsorption-induced pressure does not significantly affect the result in the given Young's modulus range (Fig. A1c). However, as you pointed out, the effect becomes noticeable at the small Young's modulus region as shown in Fig. A1b and d. The consideration of the adsorption-induced pressure limits the infinite increase of COP at low Young's modulus.

Thus, we have revised Figure 6g by the new calculation results as shown below:

Figure 6g Approximate relations between the COP and Young's modulus of the nanosponge, which were obtained at various RMPTA strains ε_d calculated using equation (35) when pore diameter (d_p) is 1.2 nm (the value in ZTC). Decreasing ε_d increases the magnitude of COP and overall nanosponge volume, making the entire system bulkier.

The revised Figure 6g actually looks almost the same as the original version. Therefore, we have separately shown the graph showing the small Young's modulus region as Supplementary Figure 9b.

Supplementary Figure 9b The relation between COP and the Young's modulus (E) of nanosponge at the very small E region.

2. When significant mechanical load is applied to a fluid-saturated porous medium, the response depends on the elastic properties of both constituents: solid and fluid. Isn't the fluid compressibility another parameter, which determines the COP? Especially that the fluid compressibility in nanopores is known to be different as compared to the macropores.

Response:

This is also an important suggestion. To investigate whether the Young's modulus of nanosponge is affected by the inclusion of H₂O, we obtained stress-strain curves with and without inclusion of H₂O by using a carbon nanotube (CNT) as a simple model, and the result is shown in Supplementary Figure 4 in the revised manuscript.

Supplementary Figure 4. Stress-strain curves of a carbon nanotube (CNT) with and without inclusion of H₂O. (a) Stress-strain curves obtained by the MD simulation. A (10,10) CNT is used and its diameter is 1.36 nm which is close to the pore width of ZTC (1.4 nm as center-to-center distance of two

pore walls). The Young's modulus of CNT is calculated as 1.2 GPa, close to that of ZTC (0.88 GPa). (b,c) Snapshots of the MD simulation at the points indicated in (a). It is found that the stress-strain curve is not significantly changed by the inclusion of H₂O.

For this simulation, a (10,10) CNT was used because its diameter (1.36 nm) is close to the pore width of ZTC (1.4 nm as center-to-center distance of two pore walls). Indeed, the calculated Young's modulus of the CNT (1.2 Ga) is close to that of ZTC (0.88 GPa). The MD simulation suggests that the inclusion of H₂O does not significantly affect the stress-strain curves, *i.e.*, Young's modulus of nanoporous media. This is because water can easily escape from the CNT and the compression mode of water is different from hydrostatic pressing. Thus, we have added the following description in the main text (page 10).

Moreover, to examine the effect of H₂O inclusion in nanopores on the mechanical property of nanosponge, a simplified model using a carbon nanotube with a diameter of 1.36 nm was investigated by the MD simulation (Supplementary Fig. 4). The calculated Young's modulus of the carbon nanotube (1.2 Ga) is close to that of ZTC (0.88 GPa). It is found that the H₂O inclusion does not significantly affect the Young's modulus of the carbon nanotube. This is because water can easily escape from the CNT and the compression mode of water is different from hydrostatic pressing.

3. Technical comments/typos:

Line 38: "is equivalent" -> "is close"

Line 115: "mechanical force of 83 MPa" -> "mechanical stress of 83 MPa"

Line 220: "imaginary nanosponge" -> "prospective nanosponge"

Figure 4(e): can you plot the force as a function of time on the same plot?

Supplementary Table 3: "...their Young's moduli are not available in literature" The moduli (pore-load moduli) of nanoporous materials can be calculated from the adsorption-induced strains, and related to the conventional Young's moduli. This has been done in a number of works recently.

Response:

Thanks for many suggestions. We have corrected the first three things. As for plotting the force as a function of time in Figure 4e (in the original manuscript), we did not measured the change of the force. We simply loaded a fixed force, and the fore is supposed to be almost constant because the deformation in this experiment is very small. For Supplementary Table 3 (in the original manuscript), we have added the data of Young's moduli in accordance with your advice, and shown as Supplementary Table 1 in the revised manuscript:

Supplementary Table 1. Bulk moduli (K) of conventional materials.

Category	Material	Pore size / nm	K^a / GPa	E^b / GPa	Reference
Metal-organic frameworks	ZIF-8	~1.1	9.2	3.0	Ito, M. et al. Chem-Eur J 19 , 13009-13016, (2013). Tan, J. C. et al. PNAS , 107 , 9938-9943 (2010).
	MOF-C30	~2	4.11	–	Han, S. S. & Goddard, W. A. J Phys Chem C 111 , 15185-15191, (2007).
Porous silicas	MOF-5	~1.5	17	2.7	Tan, J. C. & Cheetham, A. K. Chem Soc Rev 40 , 1059-1080, (2011). Bahr, D. F. & Reid, J. A. Physical Review B 76 , 184106-184112 (2007).
	SBA-15	~9	12.0	–	Kizzire, D. G. et al. Micropor Mesopor Mat 252 , 69-78, (2017).
Zeolites	FDU-12	~20	5.5	–	Mayanovic, R. A. et al. Micropor Mesopor Mat 195 , 161-166, (2014).
	Zeolite Y (FAU)	~1.2	13	37.5	Ito, M. et al. Chem-Eur J 19 , 13009-13016, (2013). Charitidis, C. A. et al., Thin Solid Films , 526 , 168-175 (2012).
	Natrolite (NAT)	0.45	48.5	77.9	Sanchez-Valle, C. et al., J. Appl. Phys. 98 , 053508 (2005).
Covalent organic frameworks	Phillipsite (PHI)	~0.6	67	–	Gatta, G. D. & Lee, Y. Micropor Mesopor Mat 105 , 239-250, (2007).
	COF-102	~2.7	21.6	~22.9	Zhou, W., Wu, H. & Yildirim, T. Chem Phys Lett 499 , 103-107, (2010).
Metals	COF-108	~2.8	4.9	~5.5	
	Al	nonporous	75.5	70	James, A. M. & Lord, M. P. (Macmillan Press, 1992).
	Ag	nonporous	103.6	83	
Au	nonporous	217	78		
Polymers	Polystyrene	nonporous	3.9-6.0	3-3.5	
	Polymethyl methacrylate	nonporous	3.7-6.3	2.4-3.4	Bondi, A. A. (Wiley, 1968).
	Polyvinyl chloride	nonporous	4.2-5.6	2.4-4.1	

^a Bulk modulus. ^b Young's modulus.

Response to the reviewer #2

Thank you very much for your precious comment. We have thoroughly revised the manuscript in accordance with your opinions, especially regarding better readability of the main text as well as simplification of too big Supplementary Information. Moreover, some detailed explanations have been inserted together with new data. The point-by-point response to your comment is shown below.

Your specific comment:

1. Although eventually understandable, the manuscript forces the reader to repeatedly shift back and forth between main text and Supplementary Information, which hampers the effective communication of the most important results. A certain familiarity with porous materials is required of the reader, which could make the text poorly readable by non-specialists.

Response:

Thanks for your suggestion of the structure of manuscript. We have simplified the Supplementary Information, and move some important description shown in the Supplementary information to the main text to improve the readability of the main text. The changed part is shown in red-colored font in the revised version.

2. The materials the authors exploit have already been published.

Response:

As you pointed out, this work is not reporting the preparation of new materials. This work is reporting a new physicochemical phenomenon, force-induced phase transition, occurring on the existing materials. We believe that a scientific paper can describe not only the development of new materials but also the finding on new phenomena. Indeed, there have been many papers describing new physical or physicochemical properties/phenomena on existing materials, for example in the fields of fullerene, carbon nanotubes, and graphene. Probably, this point was not explained well in the original manuscript. Thus, we have added the following sentence in page 2 to clearly explain the motivation of this paper in page 2.

This work is not aiming at the synthesis of new materials, but reports the new type of physicochemical methodology.

3. Most of the techniques used to quantify the claims made in the paper are standard.

Response:

Although the reviewer pointed out that most of the techniques used to quantify the claims made in the paper are standard, no one has performed force-induced reversible liquid-vapour phase transition by experiment thus far, and this is the first report. Additionally, we show the movie of the deformation of ZTC by *in situ* SEM. To the best of our knowledge, this is the first experimental demonstration of significant mechanical deformation of a single-grain of microporous material. Moreover, we have newly added the *in situ* temperature measurement up on the force-induced phase transition in Fig. 6h and Supplementary Fig. 9. We believe that these *in situ* techniques associated with the application of mechanical force on nanoporous materials are pretty new.

4. Simulations are also already published. They are never clearly explained in the text and seem to be used mainly for illustrative purposes.

Response:

The simulation results shown in this work have never been published elsewhere. In the original version, we have shown the following three simulation results:

- 1) Construction of a realistic ZTC model containing oxygen-functional groups (**Fig. 2a**)
- 2) H₂O adsorption isotherms on ZTC with and without compression (**Supplementary Fig. 3**)
- 3) MD simulation of force-induced liquid-vapour phase transition in ZTC (**Fig. 4g and h**)

All of them are new and have never been published previously by our group nor any other group. This comment may arise because of the lacking of enough explanation for each result, as the reviewer pointed out.

As for 1), we have reported a ZTC model consisting only of C and H in *Carbon* **129**, 854-862, 2018. However, to calculate adsorption isotherm of H₂O (a very polar molecule) on ZTC, the consideration of oxygen-functional groups is crucial. Thus, we have newly constructed a further realistic model of ZTC including oxygen-functional groups in this work. This is new and no one has reported such a model. Thus, we have added the following explanation in the revised manuscript (page 5).

While the previous model consists only of carbon and hydrogen atoms, a new model containing oxygen-functional groups (Fig. 2a) is constructed using the method recently developed²⁷ involving computer simulations²⁸⁻³³, and it is used for calculating adsorption isotherms of H₂O, a polar adsorbate. The CHO ratio and the compositions of oxygen-functional groups were adjusted to those reported by experiment²⁶.

For 2), the following detail explanation has been added in the main text (page 7):

It is generally anticipated for gas adsorption in nanoporous materials that the narrower the pore becomes the lower the uptake pressure becomes. However, Fig. 4d does not demonstrate such effect because of the peculiarity of H₂O adsorption into carbon materials which are intrinsically hydrophobic. To obtain deeper understanding, grand canonical Monte Carlo simulations were carried out for H₂O-adsorption isotherms in the ZTC model containing oxygen-functional groups (Supplementary Fig. 3). The simulated isotherms with and without compression shows very small difference for the H₂O uptake pressure, indicating that H₂O adsorption on ZTC is mainly governed by the strong dipole-dipole interaction between H₂O and the oxygen-containing functional groups on ZTC and the effect of the increase of the London dispersion force by pore narrowing is less effective. Additionally, at the measurement shown in Fig. 4d, the sheet-shaped sample is sandwiched by metal plates and compressed, and therefore, diffusion of H₂O vapour causes a kinetic problem, making the position of the H₂O uptake not exactly the same as the equilibrium position calculated in Supplementary Fig. 3. Also, such diffusion issue affects the presence of the hysteresis loop at low vapour pressure under compression of ZTC.

Moreover, we have added the following explanation for 3), in the main text (page 9):

When ZTC including H₂O inside its nanopores (Fig. 4g) is compressed, the liquid-density water is forcibly moved to outside of nanopores, and the expelled water is desorbed as gas (Fig. 4h). The corresponding movies showing the dynamic process at 298 K and 350 K are available in Supplementary Movie 3 and 4, respectively, showing the enhanced desorption rate at a higher temperature. The MD simulation results reasonably explain the mechanism of the proposed phase transition by mechanical force.

5. As a contradiction, a very long section is devoted to simulations in the Supplementary Information.

Response:

As the reviewer pointed out, the original version referred to Supplementary Information too much, and the main text is lacking of enough explanation on the simulation results. Thus, we have thoroughly revised the main text as well as Supplementary Information.

As we answer your comment #4, we have added the detail explanations for the three simulation results in the main text. At the same time, we have shortened the Supplementary Information. Specifically, Supplementary Figures 6, 7, 8, 9, 10 in the original manuscript have been removed, while Supplementary Figures 4 and 5 in the original manuscript are combined into Supplementary Figure 3. Moreover, the original text over 5 pages for simulation method in the Supplementary Methods was shortened into about 3 pages in the revised manuscript.

6. As a major issue, the authors do not provide a clear cut interpretation of their measurements. One may assume that what they observe is, in its main lines, related to the change in pore size (and geometry?) under compression. Now, if I am not misled, the effect of pore size on adsorption is treated already in the specialized literature. The question would rather be to explain the hysteresis loops in the adsorption/desorption isotherms. How is this related to the geometrical modification induced by the strain is however not clarified in the paper.

Response:

As the reviewer pointed out, we did not focus on the change of the hysteresis loops in the adsorption/desorption isotherms in the original manuscript. As shown in Fig. 4d, the hysteresis loop of ZTC continues to low pressure when mechanical force is applied. This is ascribed to the kinetic effect when pore size becomes smaller than the original size (1.2 nm). Thus, the following explanation has been added in the revised manuscript (page 8):

Additionally, at the measurement shown in Fig. 4d, the sheet-shaped sample is sandwiched by metal plates and compressed, and therefore, diffusion of H₂O vapour causes a kinetic problem, making the position of the H₂O uptake not exactly the same as the equilibrium position calculated in Supplementary Fig. 3. Also, such diffusion issue affects the shape of the hysteresis loop at low vapour pressure under compression of ZTC.

We found that such non-closed hysteresis loop is not observed in ethanol/GMS system, as shown in Fig. 5b. This suggests better diffusion in ethanol desorption from mesoporous GMS. Thus, the following explanation has been added in the revised manuscript (page 11):

Unlike the case of the H₂O/ZTC (Fig. 4d), the hysteresis loop closes at the same P/P₀ even under compression, indicating no serious problem for diffusion in ethanol desorption from mesoporous GMS.

7. Overall, the main purpose of the paper is left somewhat vague. Do the authors wish to contribute fundamental understanding or do they aim at proposing a new application in thermal sciences? In the former case they should focus more on the explanation of the phenomenology. If, on the other hand, their contribution is targeted to the application of soft nanosponges to heat management, they should demonstrate their cooling system in practice, e.g., by providing a proof of concept where heat transfer is actually measured.

Response:

The main purpose of this work is to contribute fundamental understanding rather than a new application. To strengthen the explanation of the phenomenology, we have made the following changes in the revised

manuscript:

(1) H₂O adsorption with and without mechanical force: As described in the answer to your comment #6, the explanation has been added as for the effect of pressing the sample on hysteresis loops of N₂ adsorption isotherms. Moreover, we have added the detail explanation on the effect of mechanical pressing on the uptake pressure of H₂O adsorption as follows in page 7:

It is generally anticipated for gas adsorption in nanoporous materials that the narrower the pore becomes the lower the uptake pressure becomes. However, Fig. 4d does not demonstrate such effect because of the peculiarity of H₂O adsorption into carbon materials which are intrinsically hydrophobic. To obtain deeper understanding, grand canonical Monte Carlo simulations were carried out for H₂O-adsorption isotherms in the ZTC model containing oxygen-functional groups (Supplementary Fig. 3). The simulated isotherms with and without compression shows very small difference for the H₂O uptake pressure, indicating that H₂O adsorption on ZTC is mainly governed by the strong dipole-dipole interaction between H₂O and the oxygen-containing functional groups on ZTC and the effect of the increase of the London dispersion force by pore narrowing is less effective. Additionally, at the measurement shown in Fig. 4d, the sheet-shaped sample is sandwiched by metal plates and compressed, and therefore, diffusion of H₂O vapour causes a kinetic problem, making the position of the H₂O uptake not exactly the same as the equilibrium position calculated in Supplementary Fig. 3. Also, such diffusion issue affects the shape of the hysteresis loop at low vapour pressure under compression of ZTC.

(2) The effect of H₂O inclusion in nanoporous materials on mechanical property: New simulation has been done by using a model carbon nanotube (Supplementary Fig. 4), and the results have been added in page 9:

When ZTC including H₂O inside its nanopores (Fig. 4g) is compressed, the liquid-density water is forcibly moved to outside of nanopores, and the expelled water is desorbed as gas (Fig. 4h). The corresponding movies showing the dynamic process at 298 K and 350 K are available in Supplementary Movie 3 and 4, respectively, showing the enhanced desorption rate at a higher temperature. The MD simulation results reasonably explain the mechanism of the proposed phase transition by mechanical force. Moreover, to examine the effect of H₂O inclusion in nanopores on the mechanical property of nanosponge, a simplified model using a carbon nanotube with a diameter of 1.36 nm was investigated by the MD simulation (Supplementary Fig. 4). The calculated Young's modulus of the carbon nanotube (1.2 Ga) is close to that of ZTC (0.88 GPa). It is found that the H₂O inclusion does not significantly affect the Young's modulus of the carbon nanotube. This is because water can easily escape from the CNT and the compression mode of water is different from hydrostatic pressing.

Supplementary Figure 4. Stress-strain curves of a carbon nanotube (CNT) with and without inclusion of H₂O. (a) Stress-strain curves obtained by the MD simulation. A (10,10) CNT is used and its diameter is 1.36 nm which is close to the pore width of ZTC (1.4 nm as center-to-center distance of two pore walls). The Young's modulus of CNT is calculated as 1.2 GPa, close to that of ZTC (0.88 GPa). (b,c) Snapshots of the MD simulation at the points indicated in (a). It is found that the stress-strain curve is not significantly changed by the inclusion of H₂O.

(3) **Adsorption-induced deformation:** The effect of the adsorption-induced deformation is analyzed and discussed in detail. The new data regarding this phenomenon has been added in Fig. 4e and Supplementary Fig. 9:

Figure 4e The volume change of ZTC induced by adsorption of H₂O, and the associated f_{ai} .

Supplementary Figure 9. Estimation of the relation between COP and E . **a** The estimation of f_0 by comparing f_{ai} obtained by experiment and f_s which can be calculated by equation (21). f_0 can be determined as the difference between f_{ai} and f_s when the deformation of nanosponge becomes 0. In the case of ZTC, f_0 is -0.18 GPa at $P/P_0=0.85$. **b** The relation between COP and the Young's modulus (E) of nanosponge at the very small E region. **c** The relation between COP and E for several different pore sizes (d_p) within 2 nm, in which water-vapour adsorption occurs in porous carbons. ϵ_d and P/P_0 are fixed to be 0.4 and 0.91, respectively.

For Fig. 4e, the following explanation has been added to strengthen the explanation of the adsorption-induced deformation as well as the adsorption-induced pressure (page 8):

From the decrease in the pore volume (31%) at $P/P_0 = 0.94$ in Fig. 4d, the strain of ZTC can be calculated as 23% under pressing with 83 MPa, and the necessary work (W_{ns}) for the deformation of the ZTC which adsorbs H_2O is therefore calculated as 18.5 kJ kg^{-1} . W_{ns} is the sum of the work (W_b) to deform blank ZTC and the additional work (W_{ai}) induced by H_2O adsorption. When nanoporous materials accommodate adsorbate inside nanopores (when adsorption occurs), adsorbed molecules generates pressure working on nanoporous materials (adsorption-induced pressure, f_{ai}), and f_{ai} causes deformation of nanoporous materials, which is known as adsorption-induced deformation³⁶⁻⁴¹. To estimate f_{ai} , the adsorption-induced deformation of ZTC was measured as shown in Fig. 4e. Since ZTC possesses an ordered structure and shows a sharp XRD peak (Fig. 2b) unlike conventional porous carbon materials, it is possible to determine the degree of shrinkage/expansion upon the H_2O adsorption by monitoring the change of the XRD peak position in situ during the H_2O adsorption experiment. By H_2O adsorption, the ZTC framework at first shrinks by 5.6% at $P/P_0 = 0.61$, whereas expands later on. At $P/P_0 = 0.95$, the ZTC framework eventually expands 2.5% compared to the original volume. Such a shrinkage-expansion behavior is similar to those found in literature, but the magnitude of deformation is far greater as a microporous material⁴¹, also demonstrating the remarkable softness of ZTC. From the volume change and the bulk modulus of ZTC, f_{ai} can be obtained as shown in Fig. 4e. By using f_{ai} , W_{ai} can be calculated according to Supplementary equation (20). For example at $P/P_0 = 0.91$, W_{ai} and W_b are 4.8 kJ kg^{-1} and 13.7 kJ kg^{-1} , respectively, showing an unignorable effect of the adsorption-induced pressure.

Moreover, it is also important to provide a proof of concept where heat transfer is actually measured, as the reviewer pointed out. Thus, we have performed additional experiment and succeeded in the detection of cooling by loading mechanical force on nanosponge. The results have been added as Fig. 7 in the revised manuscript together with some text (page 15), as shown below. The detection of cooling upon applying mechanical force has never been reported thus far, and this can be the proof of concept of this work.

To examine the feasibility of the above idea, we developed a home-made device in which both the change of gas-phase pressure and the temperature of nanosponge can be monitored during the application of mechanical force on the nanosponge (Fig. 7a). For this experiment, a mechanically tough ZTC sheet was prepared by using graphene oxide (10 wt%) as an elastic binder. To obtain a fast adsorption/desorption response, methanol was selected as a refrigerant. At beginning, the ZTC sheet

reached adsorption equilibrium under methanol vapour (17 kPa). Then, the change of temperature, as well as vapour pressure, was monitored with loading mechanical force of 30, 40, 50 and 60 MPa. As shown in Fig. 7b, the temperature change of the ZTC sheet can be observed by an infrared thermal camera through a ZnSe window. The result of the in situ monitoring is shown in Fig. 7c. By loading mechanical force, vapour pressure increases, indicating the occurrence of the force-induced evaporation of methanol from ZTC. At the same time, the temperature of the ZTC sheet clearly decreases, demonstrating cooling by mechanical compression of ZTC adsorbing methanol. Moreover, the magnitude of cooling, as well as the increase of vapour pressure, becomes larger with increasing the mechanical force, i.e., along with the increase of the ZTC deformation. Fig. 7c thus proves the most important concept of the RMPTA system: controlling temperature by phase transition generated by mechanical force applied to elastic nanosponge. On the other hand, when the external force is released, re-adsorption occurs, whereas the observed temperature rise was very small. This can be due to the measurement system which monitors the temperature change of only the edge of a ZTC sheet. When the ZTC sheet is pressed, desorption occurs mainly from the edge. On the other hand, when pressure is released, a small gap can be generated between the ZTC sheet and a pressing cylinder, and adsorption can occur the entire surface of the ZTC sheet. Thus, the temperature change of the sheet edge is less intense.

Figure 7. Temperature change by mechanical force-induced phase transition of the adsorbate. a Experimental setup. A ZTC sheet is placed in a closed chamber filled with methanol vapour (17 kPa). The temperature of the ZTC sheet can be monitored by an infrared thermal camera through a ZnSe window. The chamber is connected to an automatic adsorption analyser for controlling and monitoring vapour pressure inside the chamber. **b** Snapshot of the infrared thermal camera without (top) and with (bottom) pressing the ZTC sheet. **c** The change of methanol vapour-pressure (blue line) and temperature of ZTC (red line) when compression (30, 40, 50, and 60 MPa) and release are repeated.

Response to the reviewer #3

Thank you very much for your precious comment. We have added the data you requested and revised our manuscript in accordance with your opinions as shown below.

Your specific comment:

1. Line 119: It is not clear to me what the authors mean with “new mechanism of phase transition”? Because of the fore-induced compression the pore space cannot accommodate the condensed molecules in the pores. Thus, they are expelled, do not experience the attractive fluid-solid interactions and evaporate in a classical bulk manner. Thus, I do not see a novel mechanism here.

Response:

As you pointed out, it is quite natural to anticipate the evaporation of adsorbed molecules in nanopores when they are expelled to outside by mechanical force. Once the adsorbed molecules are expelled, they evaporate simply in a classical bulk manner. It is absolutely true. **Indeed, we have described this mechanism in the original manuscript (page 3) as shown below:**

Thus, the adsorbed refrigerant is in different equilibrium conditions from that of the bulk. Figure 1c illustrates the two different equilibrium conditions of bulk and adsorbed refrigerant. The quasi-liquid adsorbed inside nanopores is in equilibrium with the gas phase, whose pressure is much lower than the bulk saturation pressure, and therefore, its gas–liquid equilibrium line can be considered as shifting downwards. For example, at point A in Fig. 1c, the bulk refrigerant is gaseous, whereas it is condensed as quasi-liquid in nanopores. If the adsorbed refrigerant at point A is forcibly squeezed out of a nanopore, the refrigerant must obey the gas–liquid equilibrium of the bulk refrigerant (solid line in Fig. 1c), suggesting that the refrigerant at point A may evaporate immediately. Additionally, for nanoporous materials with sufficient softness and elasticity like the nanosponges discussed here, the desorbed gas can be re-adsorbed upon shape recovery of the nanosponge.

An important fact is that no one has demonstrated this phenomenon thus far, and therefore, there has been no idea of this type of phase transition up to now. This is like the egg of Columbus. Once people see how it's done, it looks quite normal and not new. Considering the concern pointed out by the reviewer, **we have added the following explanation in page 3** of the revised manuscript.

Thus, phase transition on the nanosponge is induced by the combination of the existing physicochemical phenomena, adsorption and bulk phase transition, although there has been no demonstration of this idea thus far.

Moreover, we have removed the expression such as “new phenomenon” throughout the manuscript.

2. The authors do not consider in their thermodynamic considerations the tensile Laplace pressure and sorption stresses in the confined liquids and nanopores respectively. In particular in the capillary-condensed state with concave menisci between liquid and vapour the Kelvin equation in combination with the Young-Laplace equation dictates enormous negative (tensile) pressures in the fluid for the mesoporous sponges. They result in substantial deformations, in particular contractions of nanoporous materials upon capillary condensation even without the application of external pressures (see G. Gor et al. Applied Physics Reviews 4, 011303 (2017) and references therein).

Response:

Thanks for an important suggestion. Getting the reviewer’s comment, we have considered the effect of the stress induced by confined liquids. We have indeed measured the volume change caused by H₂O adsorption in ZTC, and then calculated the adsorption-induced pressure, f_{ai} . The results have been added as Fig. 4e in the revised manuscript.

Figure 4e The volume change of ZTC induced by adsorption of H₂O, and the associated f_{ai} .

Because of the significant softness of ZTC, it shows relatively large deformation along with H₂O adsorption. The adsorption-induced pressure is not negligible, so its impact on the total mechanical force was estimated. Thus, we have added the following explanation with showing some key reference papers as follows (page 8):

From the decrease in the pore volume (31%) at $P/P_0 = 0.94$ in Fig. 4d, the strain of ZTC can be calculated as 23% under pressing with 83 MPa, and the necessary work (W_{ns}) for the deformation of the

ZTC which adsorbs H_2O is therefore calculated as 18.5 kJ kg^{-1} . W_{ns} is the sum of the work (W_b) to deform blank ZTC and the additional work (W_{ai}) induced by H_2O adsorption. When nanoporous materials accommodate adsorbate inside nanopores (when adsorption occurs), adsorbed molecules generates pressure working on nanoporous materials (adsorption-induced pressure, f_{ai}), and f_{ai} causes deformation of nanoporous materials, which is known as adsorption-induced deformation³⁶⁻⁴¹. To estimate f_{ai} , the adsorption-induced deformation of ZTC was measured as shown in Fig. 4e. Since ZTC possesses an ordered structure and shows a sharp XRD peak (Fig. 2b) unlike conventional porous carbon materials, it is possible to determine the degree of shrinkage/expansion upon the H_2O adsorption by monitoring the change of the XRD peak position in situ during the H_2O adsorption experiment. By H_2O adsorption, the ZTC framework at first shrinks by 5.6% at $P/P_0 = 0.61$, whereas expands later on. At $P/P_0 = 0.95$, the ZTC framework eventually expands 2.5% compared to the original volume. Such a shrinkage-expansion behavior is similar to those found in literature, but the magnitude of deformation is far greater as a microporous material⁴¹, also demonstrating the remarkable softness of ZTC. From the volume change and the bulk modulus of ZTC, f_{ai} can be obtained as shown in Fig. 4e. By using f_{ai} , W_{ai} can be calculated according to Supplementary equation (20). For example at $P/P_0 = 0.91$, W_{ai} and W_b are 4.8 kJ kg^{-1} and 13.7 kJ kg^{-1} , respectively, showing an unignorable effect of the adsorption-induced pressure.

Newly added references:

36. Meehan, F. T. The expansion of charcoal on sorption of carbon dioxide. *Proc. R. Soc. London, Ser. A* **115**, 199-207 (1927)
37. Gor, G. Y. & Neimark, A. V. Adsorption-induced deformation of mesoporous solids. *Langmuir* **26**, 13021-13027 (2010).
38. Gor, G. Y. *et al.* Elastic response of mesoporous silicon to capillary pressures in the pores. *Appl. Phys. Lett.* **106**, 261901-261905 (2015).
39. Balzer, C., Cimino, R. T., Gor, G. Y., Neimerk, A. V. & Reichenauer, G. Deformation of microporous carbons during N_2 , Ar, and CO_2 adsorption: Insight from the density functional theory. *Langmuir* **32**, 8265-8274 (2016).
40. Gor, G. Y. & Bernstein, N. Revisiting Bangham's law of adsorption-induced deformation: changes of surface energy and surface stress. *Phys. Chem. Chem. Phys.* **18**, 9788-9798 (2016).
41. Gor, G. Y., Huber, P. & Bernstein, N. Adsorption-induced deformation of nanoporous materials-A review. *Appl. Phys. Rev.* **4**, 11303-11324 (2017).

3. In the case of the application of soft porous materials with low Young modulus, suggested by the authors as a way to increase the coefficient of performance, the sorption-deformations mentioned above will result in substantial nanopore contractions and thus in a spontaneous fluid-adsorption-induced reduction of pore space - even without external forces. Thus, the overall porosity available for fluid adsorption will be substantially reduced which may even overbalance the increased contraction response to external pressure and thus decrease the COP.

Response:

Getting the reviewer’s comment, we have revised the calculation of COP to include the effect of the adsorption-induced pressure. Please see the details in the revised Supplementary Methods. The major results are summarized in the following Figure A1.

Figure A1 The relation between COP and Young’s modulus. (a,b) The data shown in the original manuscript. (c,d) The revised data considering the effect of the adsorption-induced pressure. (b) and (d)

show the small Young's modulus regions of (a) and (c), respectively.

Figure A1a is the graph which was shown in the original manuscript. The consideration of the adsorption-induced pressure does not significantly affect the result in the given Young's modulus range (Fig. A1c). However, as you concerned, COP is decreased at the small Young's modulus region (< 0.4 GPa) as shown in Fig. A1b and d.

Thus, we have revised Figure 6g by the new calculation results as shown below:

Figure 6g Approximate relations between the COP and Young's modulus of the nanosponge, which were obtained at various RMPTA strains ε_d calculated using equation (35) when pore diameter (d_p) is 1.2 nm (the value in ZTC). Decreasing ε_d increases the magnitude of COP and overall nanosponge volume, making the entire system bulkier.

The revised Figure 6g actually looks almost the same as the original version. Therefore, we have separately shown the graph showing the small Young's modulus region as Supplementary Figure 9b.

Supplementary Figure 9b The relation between COP and the Young's modulus (E) of nanosponge at the very small E region.

4. The authors discuss the refrigeration cycles theoretically only, but do not show any experimental data actually proofing their cooling/heating concept.

Response:

We consider this suggestion as important, and have performed additional experiment to directly monitor the temperature change upon loading mechanical force on elastic nanoporous material with the presence of methanol vapour. **The results have been added as Fig. 7** in the revised manuscript, as shown below. By applying mechanical force on ZTC which adsorbs methanol, desorption of methanol is observed, and at the same time, temperature decrease can be successfully detected. The detection of cooling upon applying mechanical force has never been reported thus far, and this is the first demonstration of the cooling induced by mechanical force. On the other hand, when the external force is released, re-adsorption occurs, whereas temperature rise was very small. This can be due to the measurement system which monitors the temperature change of only the edge of a ZTC sheet. When the ZTC sheet is pressed, desorption occurs mainly from the edge. On the other hand, when pressure is released, a small gap can be generated between the ZTC sheet and a pressing cylinder, and adsorption can occur the entire surface of the ZTC sheet. Thus, the temperature change of the sheet edge is less intense. **We have added these results in the revised manuscript (page 15).**

To examine the feasibility of the above idea, we developed a home-made device in which both the change of gas-phase pressure and the temperature of nanosponge can be monitored during the application of mechanical force on the nanosponge (Fig. 7a). For this experiment, a mechanically tough ZTC sheet was prepared by using graphene oxide (10 wt%) as an elastic binder. To obtain a fast adsorption/desorption response, methanol was selected as a refrigerant. At beginning, the ZTC sheet reached adsorption equilibrium under methanol vapour (17 kPa). Then, the change of temperature, as well as vapour pressure, was monitored with loading mechanical force of 30, 40, 50 and 60 MPa. As shown in Fig. 7b, the temperature change of the ZTC sheet can be observed by an infrared thermal camera through a ZnSe window. The result of the in situ monitoring is shown in Fig. 7c. By loading mechanical force, vapour pressure increases, indicating the occurrence of the force-induced evaporation of methanol from ZTC. At the same time, the temperature of the ZTC sheet clearly decreases, demonstrating cooling by mechanical compression of ZTC adsorbing methanol. Moreover, the magnitude of cooling, as well as the increase of vapour pressure, becomes larger with increasing the mechanical force, i.e., along with the increase of the ZTC deformation. Fig. 7c thus proves the most important concept of the RMPTA system: controlling temperature by phase transition generated by mechanical force applied to elastic nanosponge. On the other hand, when the external force is released,

re-adsorption occurs, whereas the observed temperature rise was very small. This can be due to the measurement system which monitors the temperature change of only the edge of a ZTC sheet. When the ZTC sheet is pressed, desorption occurs mainly from the edge. On the other hand, when pressure is released, a small gap can be generated between the ZTC sheet and a pressing cylinder; and adsorption can occur the entire surface of the ZTC sheet. Thus, the temperature change of the sheet edge is less intense.

Figure 7. Temperature change by mechanical force-induced phase transition of the adsorbate. a *Experimental setup. A ZTC sheet is placed in a closed chamber filled with methanol vapour (17 kPa). The temperature of the ZTC sheet can be monitored by an infrared thermal camera through a ZnSe window. The chamber is connected to an automatic adsorption analyser for controlling and monitoring vapour pressure inside the chamber. b* *Snapshot of the infrared thermal camera without (top) and with (bottom) pressing the ZTC sheet. c* *The change of methanol vapour-pressure (blue line) and temperature of ZTC (red line) when compression (30, 40, 50, and 60 MPa) and release are repeated.*

REVIEWERS' COMMENTS:

Reviewer #1 (Remarks to the Author):

In my understanding the initial version of the manuscript was already very convincing. The only detail which was missing in the discussion (and in estimation of COP) were the "adsorption-induced deformation" effects. I was pleased to see that one of the other reviewers made exactly the same point. After the authors addressed this issue (and other minor issues) I do not see any obstacles to recommend it for publication.

I believe that this paper will make a significant impact, and I recommend its publication in the current form.

Reviewer #2 (Remarks to the Author):

I have read the rebuttal letter by the authors and the new version of paper and supplementary informations. After some thinking, I decided to stick to my original position that was already explained in detail in my previous report.

I am overall convinced that the paper has good scientific quality and that it may be interesting for the community working on porous materials and their applications. However, despite the clarifications provided by the authors and the details added to the text, I still did not perceive the ground breaking novelty of the results presented in the paper. As concerning this crucial aspect, the authors clarify their intention to communicate a new physicochemical phenomenon. Elsewhere they state instead that we have removed the expression such as "new phenomenon" throughout the manuscript. In the present referee's view this kind of apparent self-contradiction is a clear manifestation of the very issue that has been perplexing me a lot: apart from technical aspects, what is such an important novelty expected to lead to ground-breaking advancement in the understanding of the wetting behavior of porous, in this case also elastic, materials to be communicated in a wide scope journal devoted to reporting substantial advancements in different fields of science? In fact, after reading the new version of the paper, still no substantially new phenomenon of widespread interest seems to emerge, as the authors themselves are implicitly acknowledging. As I already had the chance to say, the combination of material properties and known phenomena occurring in nanoporous systems can be exploited for innovative applications. However, answering to my precise question, the authors explicitly state that they are not interested

in communicating new technological ideas which, at any rate, would have required additional details to provide evidence of the proposal effectiveness.

In these conditions, after long meditation and with some regret, I am not in the position of recommending the paper for publication in Nature Communications, given the wide scope of the journal and its mission toward communicating most significant advances in science. On the other hand, I remain convinced that the authors work is well done, interesting and certainly worth being communicated to the specialized readership interested in porous materials and their applications. I hence suggest submission of the present paper to a journal with more focused scope, which may be done, in my opinion, by substantially keeping the current version of the manuscript.

Reviewer #3 (Remarks to the Author):

The authors addressed all of my criticisms and suggestions of the first reviewing round. I recommend a publication of the revised version of the manuscript.

Response to the reviewer #1

Provided comment:

In my understanding the initial version of the manuscript was already very convincing. The only detail which was missing in the discussion (and in estimation of COP) were the "adsorption-induced deformation" effects. I was pleased to see that one of the other reviewers made exactly the same point. After the authors addressed this issue (and other minor issues) I do not see any obstacles to recommend it for publication.

I believe that this paper will make a significant impact, and I recommend its publication in the current form.

Response:

Thank you very much for your kind recheck of our revised manuscript. Thanks to your precious comment, our manuscript has been greatly improved. We also hope that this paper will make a significant impact.

Response to the reviewer #2

Provided comment:

I have read the rebuttal letter by the authors and the new version of paper and supplementary informations. After some thinking, I decided to stick to my original position that was already explained in detail in my previous report.

I am overall convinced that the paper has good scientific quality and that it may be interesting for the community working on porous materials and their applications. However, despite the clarifications provided by the authors and the details added to the text, I still did not perceive the ground breaking novelty of the results presented in the paper. As concerning this crucial aspect, the authors clarify their intention to communicate a new physicochemical phenomenon. Elsewhere they state instead that we have removed the expression such as "new phenomenon" throughout the manuscript. In the present referee's view this kind of apparent self-contradiction is a clear manifestation of the very issue that has been perplexing me a lot: apart from technical aspects, what is such an important novelty expected to lead to ground-breaking advancement in the understanding of the wetting behavior of porous, in this case also elastic, materials to be communicated in a wide scope journal devoted to reporting substantial advancements in different

fields of science? In fact, after reading the new version of the paper, still no substantially new phenomenon of widespread interest seems to emerge, as the authors themselves are implicitly acknowledging. As I already had the chance to say, the combination of material properties and known phenomena occurring in nanoporous systems can be exploited for innovative applications. However, answering to my precise question, the authors explicitly state that they are not interested in communicating new technological ideas which, at any rate, would have required additional details to provide evidence of the proposal effectiveness.

In these conditions, after long meditation and with some regret, I am not in the position of recommending the paper for publication in Nature Communications, given the wide scope of the journal and its mission toward communicating most significant advances in science. On the other hand, I remain convinced that the authors work is well done, interesting and certainly worth being communicated to the specialized readership interested in porous materials and their applications. I hence suggest submission of the present paper to a journal with more focused scope, which may be done, in my opinion, by substantially keeping the current version of the manuscript.

Response:

We seriously consider your comment regarding the consistency of the context throughout the text. We have agreed that the proposed method cannot be said to deliver “new” physicochemical phenomenon, and the phase-transition reported here is the combination of existing physicochemical phenomenon. We have carefully checked the text again and revised the expressions including any small nuance which are related to your concern. Specifically, the following sentence has been removed: “Conventionally, only two methods are used to convert adsorbed liquid into gas: heating the adsorbent or reducing the gas-phase pressure. The proposed force-driven liquid–gas phase transition thus demonstrate another method of adsorbent phase conversion.”

On the other hand, we believe that this work contains significant novelty and new ideas of functions of nanoporous materials, as other two reviewers appreciate this work and agree with the publication in Nature Communications.

Response to the reviewer #3

Provided comment:

The authors addressed all of my criticisms and suggestions of the first reviewing round. I recommend a publication of the revised version of the manuscript.

Response:

Thank you very much for your kind recheck of our revised manuscript. We deeply appreciate your expert comment regarding adsorption-induced deformation and the occurrence of internal force. By taking this important effect into consideration, the scientific quality of this work could be greatly improved. We hope that this paper will not only make a significant impact but also contribute to the science community of nanoporous materials and adsorption.